# Scc2 counteracts a Wapl-independent mechanism that releases cohesin from chromosomes during G1

Madhusudhan Srinivasan[1]*, Naomi J Petela[1], Johanna C Scheinost[1], James Collier[1], Menelaos Voulgaris[1], Maurici B Roig[1], Frederic Beckouët[2], Bin Hu[3], Kim A Nasmyth[1]*

[1]Department of Biochemistry, University of Oxford, Oxford, United Kingdom; [2]Laboratoire de Biologie Moléculaire Eucaryote, Centre de Biologie Intégrative (CBI), Université de Toulouse, Toulouse, France; [3]Department of Molecular Biology and Biotechnology, University of Sheffield, Sheffield, United Kingdom

**Abstract** Cohesin's association with chromosomes is determined by loading dependent on the Scc2/4 complex and release due to Wapl. We show here that Scc2 also actively maintains cohesin on chromosomes during G1 in *S. cerevisiae* cells. It does so by blocking a Wapl-independent release reaction that requires opening the cohesin ring at its Smc3/Scc1 interface as well as the D loop of Smc1's ATPase. The Wapl-independent release mechanism is switched off as cells activate Cdk1 and enter G2/M and cannot be turned back on without cohesin's dissociation from chromosomes. The latter phenomenon enabled us to show that in the absence of release mechanisms, cohesin rings that have already captured DNA in a Scc2-dependent manner before replication no longer require Scc2 to capture sister DNAs during S phase.
DOI: https://doi.org/10.7554/eLife.44736.001

*For correspondence:
madhusudhan.srinivasan@bioch.
ox.ac.uk (MS);
ashley.nasmyth@bioch.ox.ac.uk
(KAN)

Competing interests: The authors declare that no competing interests exist.

## Introduction

Accurate chromosome segregation is only possible because monumental topological problems posed by the sheer size and physical properties of DNA are overcome by highly conserved DNA motors, namely condensin and cohesin. Replicated DNA is weaved into discrete chromatids during mitosis by condensin (*Hirano et al., 1997*) while sister chromatids are held together by cohesin (*Guacci et al., 1997*; *Michaelis et al., 1997*), which is essential for their bi-orientation on mitotic spindles.

Both complexes contain a pair of rod-shaped Smc proteins (Smc1/3 in cohesin) whose association via their hinge domains creates V-shaped heterodimers with ATPase domains at their vertices. These are interconnected by kleisin subunits to form trimeric rings, whose activity is regulated by a set of hook-shaped proteins composed of HEAT repeats known as HAWKs (HEAT repeat proteins Associated With Kleisins) (*Wells et al., 2017*). Regulation by HAWKs distinguishes cohesin and condensin from bacterial Smc/kleisin complexes and the eukaryotic Smc5/6 complex, whose kleisin subunits associate instead with tandem-winged helical domain proteins called KITEs (*Palecek and Gruber, 2015*). Cohesin has three HAWKs: Scc3 is permanently bound while Scc2/Nipbl and Pds5 appear interchangeable (*Petela et al., 2018*).

Condensin has the remarkable ability to form and expand in a processive manner DNA loops in vitro (*Ganji et al., 2018*), an activity known as loop extrusion, previously postulated to explain how condensin transforms interphase chromosomes into thread-like chromatids while at the same time accumulating along their longitudinal axes (*Goloborodko et al., 2016*; *Nasmyth, 2001*; *Naumova et al., 2013*). Cohesin has more diverse activities. In addition to its canonical role of

holding together sister chromatids, cohesin also organises interphase chromatin into defined territories called TADs (Topologically Associated Domains), a process also thought to be driven by loop extrusion (*Fudenberg et al., 2016*; *Rao et al., 2017*; *Sanborn et al., 2015*). Cohesin is thought to mediate cohesion by entrapping sister DNAs inside its tripartite ring (*Srinivasan et al., 2018*). It can also entrap individual DNAs prior to DNA replication and this may be a feature of its chromosomal association throughout the cell cycle. Whether DNAs are entrapped within cohesin rings during loop extrusion is not known.

Loading of cohesin onto chromosomes as well as entrapment of mini-chromosome DNAs by cohesin rings depends on both Scc3 and Scc2 but not on Pds5. Loading also requires Scc4, which binds to an unstructured N-terminal domain within Scc2. Because neither Scc2 nor Scc4 are required to maintain cohesion following S phase, Scc2/4 has long been thought to function merely as a 'loading complex'(*Ciosk et al., 2000*). However, the finding that Scc2 associates with chromosomal cohesin long after loading (*Rhodes et al., 2017a*) suggests that this may not be the whole story. Because it is essential for activating cohesin's ATPase (*Petela et al., 2018*), Scc2 may have a key role in activating the DNA translocase activity that powers loop extrusion.

DNAs are released from cohesin rings by two mechanisms, either through kleisin cleavage by separase, which occurs at the onset of anaphase (*Uhlmann et al., 1999*), or at other stages of the cell cycle via a separase-independent mechanism that requires the binding to Pds5 and Scc3 of a fourth regulatory subunit called Wapl. Wapl-dependent releasing activity (RA) is abrogated by mutations on Pds5 and Scc3 (*pds5S81R* and *scc3K404E*) that abolish their interaction with Wapl (*Beckouët et al., 2016*; *Chan et al., 2012*) (*Figure 1A*). Wapl-dependent RA induces disengagement of the ring's Smc3/kleisin interface, thereby creating a gate through which DNAs can escape (*Beckouët et al., 2016*; *Murayama and Uhlmann, 2015*). The steady state level of chromatin associated cohesin during G1 is therefore determined by the rates of Scc2 catalysed loading and Wapl-dependent RA (*Figure 1A*). Because it would destroy sister chromatid cohesion, Wapl-dependent RA is neutralised during DNA replication through acetylation of Smc3 K112 and K113 by Eco1 (*Rolef Ben-Shahar et al., 2008*; *Ivanov et al., 2002*; *Unal et al., 2008*). Though normally essential for cell viability, Eco1 is dispensable in mutants defective in release (*Rolef Ben-Shahar et al., 2008*; *Chan et al., 2012*; *Rowland et al., 2009*; *Srinivasan et al., 2018*; *Sutani et al., 2009*).

Both cohesion establishment and Smc3 acetylation are tightly coupled to DNA replication. For example, cohesin that loads onto chromosomes during G2 cannot connect sister DNAs (*Haering et al., 2004*) and mutation of non-essential fork-associated proteins (Ctf4, Ctf18, Tof1, Csm3, Mrc1and Chl1) causes cohesion defects without adversely affecting replication (*Borges et al., 2013*; *Zheng et al., 2018*). Nevertheless, the mechanism by which cohesion is established remains poorly understood. Photo-bleaching experiments showing that replication fork passage does not per se induce cohesin's dissociation (*Rhodes et al., 2017b*) suggest either that replication forks actually pass through cohesin rings that have entrapped DNA ahead of the fork or that such cohesin rings are opened transiently as they are passed on to, and subsequently entrap sisters.

One way of providing insight into cohesion establishment would be to determine whether it depends on Scc2. Replication through rings should not require a second Scc2-dependent loading reaction while ring opening and re-loading on lagging or leading (or both) strands would be expected to do so. The recent observation that cohesin loaded onto a double stranded DNA in vitro is capable of capturing a second single stranded DNA molecule in a manner dependent on Scc2 (*Murayama et al., 2018*) raises the possibility that cohesion is established by rings associated with a leading strand capturing the lagging strand through Scc2 catalysed ring opening. If so, Scc2 must be required during replication itself as well for loading cohesin onto un-replicated chromatin.

We therefore set out to answer the following simple question: In cells lacking Wapl-dependent RA, is Scc2 required to build cohesion during S phase in cells that have already loaded cohesin onto chromosomes during G1? To answer this, we set out to inactivate Scc2 in late G1 cells that had already loaded cohesin onto chromosomes, while Scc2 was still active, and then allow these cells to undergo S phase in the absence of any further Scc2 activity. Unexpectedly, inactivation of Scc2 in pre-replicative cells causes cohesin unloading throughout the genome, even in cells that lack Wapl-dependent RA, a phenomenon that precluded our intended experiment.

We subsequently discovered that release in G1 cells involving disengagement of the Smc3/Scc1 interface is in fact a Wapl-independent process that is actively blocked by Scc2. Scc2 is therefore not merely a loader and Wapl is not an intrinsic aspect of releasing activity (RA). Wapl-independent RA

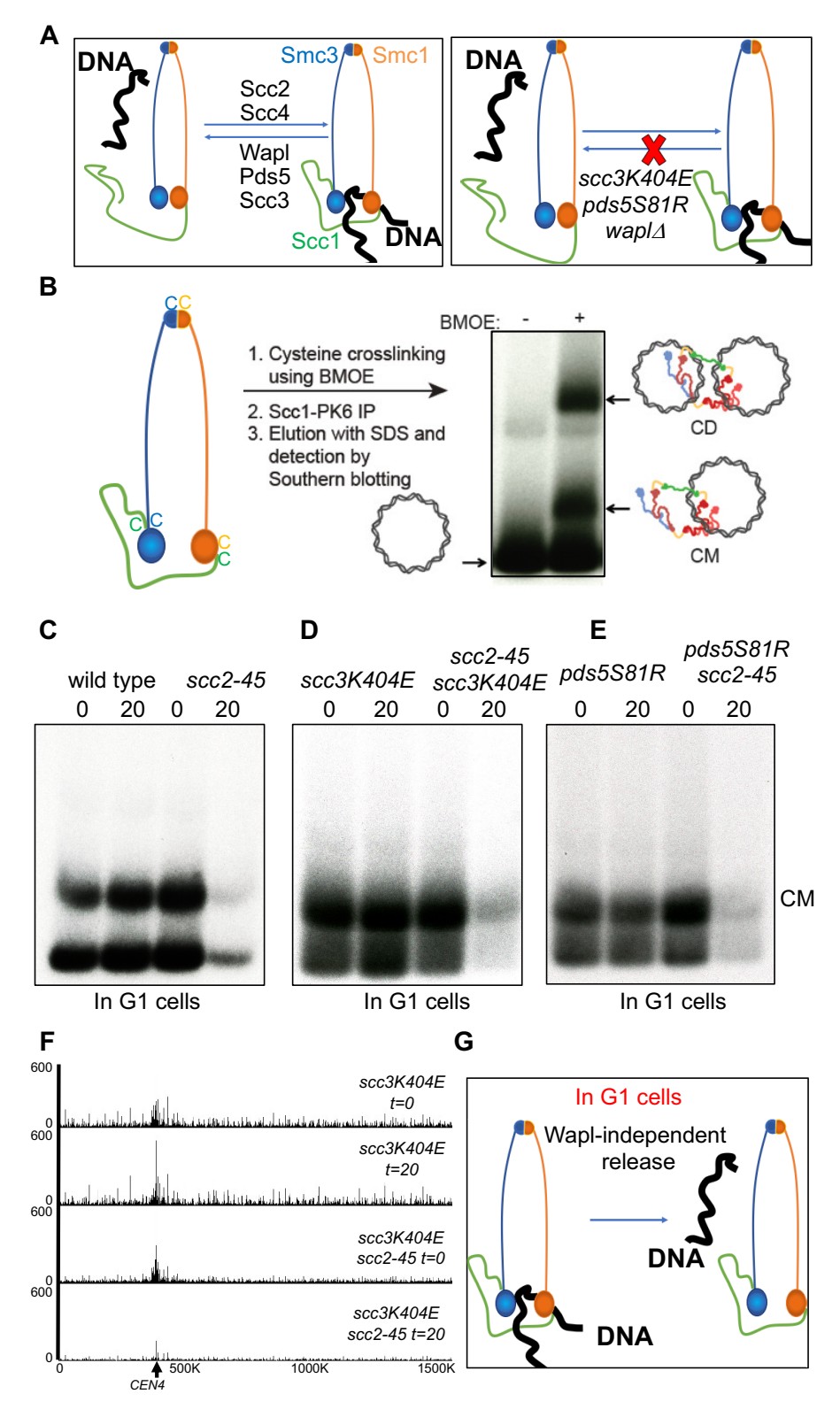

**Figure 1.** A Wapl-independent activity releases cohesin from chromosomes in G1 cells. (**A**) Cohesin's association with DNA is regulated by two opposing activities: Scc2-Scc4 complex loads cohesin onto DNA, while Pds5, Scc3 and Wapl constitute the releasing activity that releases cohesin from DNA by opening the Smc3-Scc1 interface. Mutations in Scc3 (*scc3K404E*) Pds5 (*pds5S81R*) and deletion of the *WAPL* gene abrogate Wapl mediated releasing activity and lead to cohesin's stable association with DNA. (**B**) Schematic of the mini-chromosome IP assay: 6C strain (K23889) with cysteine

*Figure 1 continued on next page*

*Figure 1 continued*

pairs at all three ring subunit interfaces (2C Smc3: E570C S1043C, 2C Smc1: G22C K639C and 2C Scc1 C56 A547C) and carrying a 2.3 kb circular mini-chromosome was subjected to in vivo crosslinking with BMOE. DNAs associated with cohesin immune-precipitates (Scc1-PK6) were denatured with SDS and separated by agarose gel electrophoresis. Southern blotting reveals two forms of DNA unique to cells treated with BMOE: CMs (cohesin entrapping individual mini-chromosomes) and CDs (cohesin entrapping a pair of sister mini-chromosomes). (C) WT (K23972) and *scc2-45* (K25238) 6C strains were arrested in late G1 by overexpression of nondegradable Sic1 at 25°C as described in Materials and Methods. The cultures were shifted to 37°C for 20 min, aliquots drawn before (0) and after (20) temperature shift (to inactivate Scc2) were subjected to mini-chromosome IP. (D) *scc3K404E* (K25313) and *scc3K404E scc2-45* (K25316) 6C strains were arrested in late G1. The cultures were shifted to 37°C for 20 min, aliquots drawn before (0) and after (20) temperature shift (to inactivate Scc2) were subjected to mini-chromosome IP. Also see S1B. (E) *pds5S81R* (K25311) and *pds5S81R scc2-45* (K25312) 6C strains were arrested in late G1. The cultures were shifted to 37°C for 20 min, aliquots drawn before (0) and after (20) temperature shift (to inactivate Scc2) were subjected to mini-chromosome IP. (F) *scc3K404E* (K25313) and *scc3K404E scc2-45* (K25316) strains were arrested in late G1. The cultures were shifted to 37°C for 20 min, aliquots drawn before (0) and after (20) temperature shift (to inactivate Scc2) were analysed by Calibrated-ChIP-sequencing (Scc1-PK). Cohesin profile along chromosome four is shown. Also see *Figure 1—figure supplement 1*. (G) Even in cells that lack Wapl mediated releasing activity, inactivation of Scc2 in G1 cells leads to release of DNA entrapped within cohesin rings. This suggests that an activity that is Wapl-independent is capable of releasing cohesin from DNA.

DOI: https://doi.org/10.7554/eLife.44736.002

The following figure supplement is available for figure 1:

**Figure supplement 1.** A Wapl-independent activity releases cohesin from chromosomes in G1 cells.
DOI: https://doi.org/10.7554/eLife.44736.003

is switched off as cells undergo S phase in a manner that does not require either replication or acetylation of Smc3 by Eco1 but involves Cdk1. Because it cannot be switched back on (upon Cdk1's subsequent inactivation) without cohesin's removal from chromosomes, we were able to show that cohesin loaded prior to replication can create cohesion without Scc2, a finding that has profound implications regarding the mechanism of cohesion establishment.

## Results

### A Wapl-independent activity releases cohesin from chromosomes in G1 cells

To measure DNA entrapment within cohesin rings, we used budding yeast cells containing cysteine pairs at all of cohesin ring's three interfaces (6C) that can be cross-linked by the homobifunctional crosslinker BMOE (*Gligoris et al., 2014*). Following SDS treatment, gel electrophoresis and Southern blotting reveals two types of circular mini-chromosomes associated with 6C cohesin only when cells are treated with BMOE: catanated monomers (CMs) that correspond to monomeric supercoiled DNAs catenated by a single cohesin ring and catanated dimers (CDs) that correspond to sister DNAs catenated by a single ring (*Figure 1B*) (*Srinivasan et al., 2018*). CMs are formed when cohesin loads onto chromosomes either before or after (see below) DNA replication while CDs only form when cohesin builds cohesion during S phase.

To ascertain whether cohesin's 'loading complex' is required to establish cohesion during S phase, we set out to determine whether its Scc2 subunit is required to convert CMs formed during G1 into CDs during replication. To do this, we first established a protocol for inactivating Scc2 in G1 cells that accumulate Scc1 to high levels and load cohesin onto chromosomes. We therefore arrested wild type (WT) and *scc2-45* (a temperature sensitive *SCC2* allele) 6C cells in late G1 by expression of a non-degradable form of the Cdk1 inhibitor Sic1 at the permissive temperature (25°C). Cohesin that accumulates under these conditions forms CMs (*Srinivasan et al., 2018*), which disappear in *scc2-45* but not *SCC2* (WT) cells when they are shifted to 37°C for 20 min (*Figure 1C*).

The simplest explanation for cohesin's dissociation from mini-chromosomes upon Scc2's inactivation is that the latter is required to re-load chromosomal cohesin that is continually turning over due to Wapl-dependent RA (*Figure 1A*), which is active in these cells (*Chan et al., 2012*; *Lopez-Serra et al., 2013*). To test this, we repeated the experiment using cells carrying *scc3K404E* (*Figure 1D* and *Figure 1—figure supplement 1B*) or *pds5S81R* (*Figure 1E*) mutations that abolish Wapl-dependent RA in otherwise wild type cells (*Beckouët et al., 2016*). Surprisingly, neither mutation abrogated loss of CMs in *scc2-45* cells (*Figure 1D and E*). Calibrated ChIP-seq confirmed that even when Wapl mediated release is abrogated, inactivation of Scc2 leads to dissociation of about

80% of chromosome associated cohesin (*Figure 1F* and *Figure 1—figure supplement 1A*). This suggests that in late G1 cells, a Wapl-independent activity releases cohesin from chromosomes (*Figure 1G*).

Because the notion that cohesin in G1 is released by a Wapl-independent mechanism is striking and unprecedented, it was necessary to ensure that Wapl-dependent RA really was inactive in the experiments described above. However, when we tried to repeat these experiments in cells lacking Wapl itself, we discovered that *WPL1* deletion adversely affects the proliferation of *scc2-45* cells. Fortunately, this synthetic lethality is relieved by a point mutation in *SMC3* (*smc3R1008I*) that has little or no phenotype on its own and importantly does not restore releasing activity as it enhances not reduces proliferation of *wpl1Δ eco1Δ* cells (*Figure 1—figure supplement 1C*). We do not fully understand the mechanism by which *smc3R1008I* influences cohesin (Hu et al., in preparation), the mutation nevertheless permitted us to assess the effect of inactivating *scc2-45* in cells completely lacking Wapl.

Calibrated ChIP-seq showed that shifting late G1 arrested cells to 37°C for 20 min had no effect on chromosomal cohesin in *smc3R1008I wpl1Δ* cells but caused a major reduction throughout the genome in *smc3R1008I wpl1Δ scc2-45* cells (*Figure 2A* and *Figure 1—figure supplement 1D*). *Figure 2C* documents this effect by plotting the ratio of average cohesin levels 60 kb either side of all 16 centromeres before and 20 min after the temperature shift (*Figure 2C*). The key point is that inactivation of *scc2-45* causes about a sevenfold reduction in chromosomal cohesin, confirming that the activity which releases cohesin from DNA in G1 upon Scc2 inactivation is truly independent of Wapl.

## The Wapl-independent RA is active only in G1 and not G2

Remarkably, inactivation of *scc2-45* had no effect on chromosomal cohesin in cells arrested in G2/M phase by nocodazole. Thus, neither CMs nor CDs (*Figure 1—figure supplement 1B and F*) were altered by shifting post-replicative *scc3K404E scc2-45* cells to 37°C. Likewise, the calibrated ChIP-seq profiles and −60 kb to +60 kb temperature shift ratio profiles of *wpl1Δ SCC2* and *wpl1Δ scc2-45* G2/M phase cells were indistinguishable (*Figure 2B and D* and *Figure 1—figure supplement 1E*). This is consistent with the previous finding that Scc2 is not required to maintain cohesion during G2/M (*Ciosk et al., 2000*).

To exclude the possibility that the phenomenon is a peculiarity of the *scc2-45* allele, we repeated the experiment with an Auxin degron allele (*SCC2-3XmAID*) (*Figure 1—figure supplement 1H and I*). Cohesin ChIP profiles show that Scc2 depletion induced by addition of synthetic auxin (Indole-3-acetic acid) for 60 min caused a drastic reduction in cohesin levels on chromosome four in G1 but not G2 arrested cells (*Figure 2E and F*). Plotting the −60 kb to +60 kb ratio profiles before and after auxin addition again revealed a major discrepancy between *SCC2* (blue) and *SCC2-3XmAID* (red) in G1 (*Figure 2G*) but not in G2 (*Figure 2H*) *wpl1Δ* cells. Thus, Scc2 depletion also causes cohesin's dissociation from G1 but not G2 chromatin genome wide.

Because Scc2's inactivation is not accompanied by any change in the overall levels of Scc1 (*Figure 1—figure supplement 1G*), we conclude that Scc2 is essential to maintain cohesin's association with chromosomes during late G1 even when there is no turnover. This implies that G1 but not G2 cohesin has the ability to dissociate from chromatin through a Wapl-independent mechanism (*Figure 2I and J*) and that Scc2 is required to counteract this activity. Three important corollaries follow. First, Scc2 is not merely involved in loading cohesin onto chromosomes but acts long afterwards to prevent release. Second, contrary to prevailing wisdom, Wapl is in fact not essential for cohesin to dissociate from chromosomes. Third, this Wapl-independent release mechanism is cell cycle regulated and turned off in G2 cells, with the result that Scc2 is no longer required to maintain cohesin on chromosomes after replication.

## Scc2 does not require Scc4 to block Wapl-independent RA

How does Scc2 block release? Our first step was to investigate the role of Scc4, which interacts with the N-terminus of Scc2 (*Hinshaw et al., 2015*). It has been suggested Scc4 is necessary for Scc2's stability (*Watrin et al., 2006*). However, we found that inactivation of Scc4 by shifting *scc4-4* cells to the restrictive temperature does not in fact affect Scc2 levels in late G1 cells (*Figure 3—figure supplement 1A*). Thus, inactivating Scc4 using *Scc4-4* permitted us to ask whether Scc2 can still inhibit

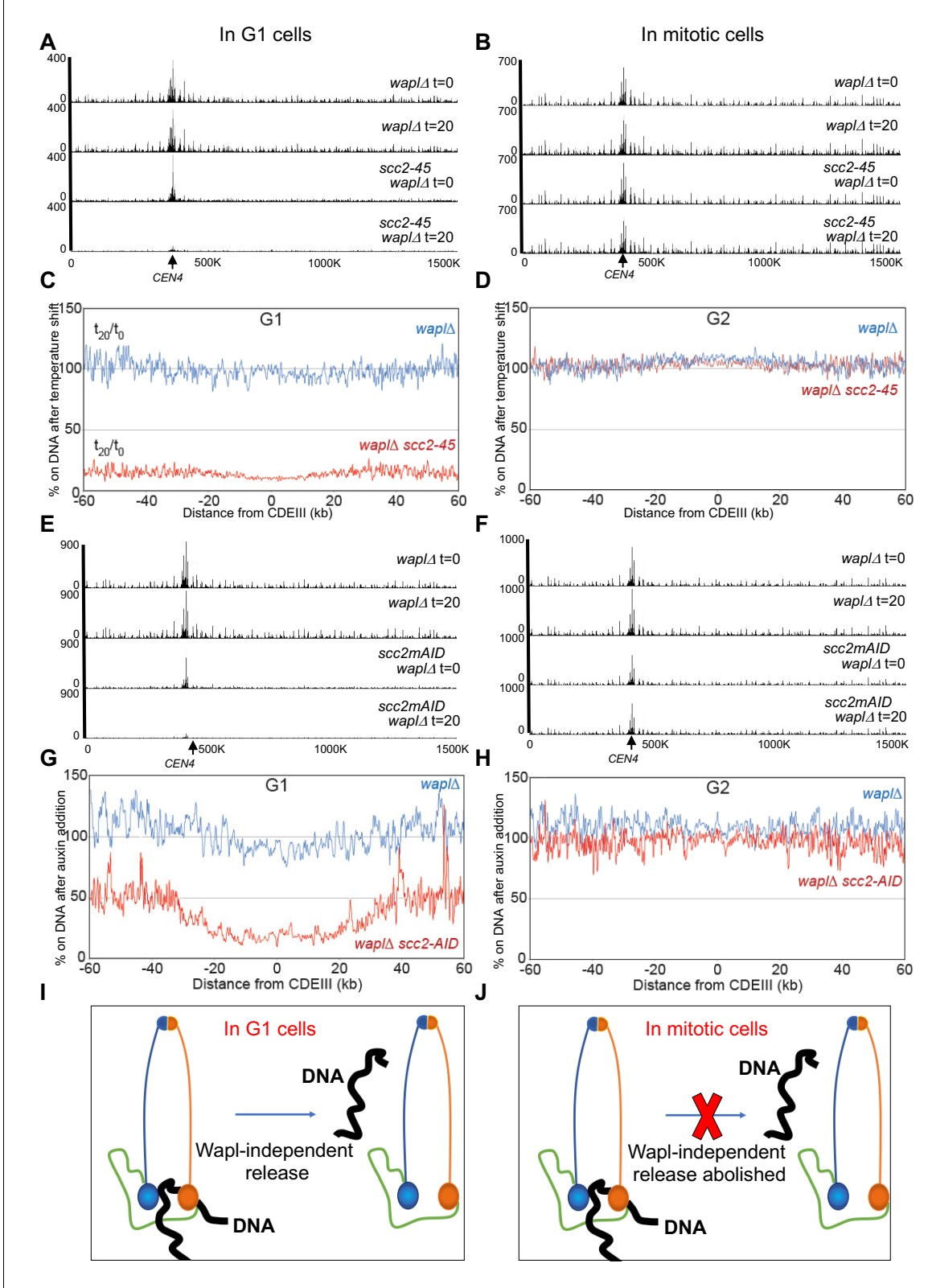

**Figure 2.** The Wapl-independent activity that releases cohesin from chromosomes is active only in G1 cells. (**A**) *waplΔ* (K22296) and *waplΔ scc2-45* (K22294) strains were arrested in late G1 at 25°C and subjected to temperature shift to 37°C for 20 min. 0- and 20 min samples were analysed by calibrated ChIP-sequencing (Scc1-PK6) as detailed in Materials and Methods. Cohesin ChIP profiles along chromosomes four is shown. Also see *Figure 1—figure supplement 1D*. (**B**) *waplΔ* (K22296) and *waplΔ scc2-45* (K22294) strains were arrested in G2 with nocadazole at 25°C and subjected to

*Figure 2 continued on next page*

*Figure 2 continued*

temperature shift to 37°C for 20 min. 0- and 20 min samples were analysed by calibrated ChIP-sequencing (Scc1-PK6). Cohesin ChIP profiles along chromosomes four is shown. Also see *Figure 1—figure supplement 1E*. (**C**) Data form (**A**) is plotted to show the ratio of average cohesin levels 60 kb on either side of all 16 centromeres before and 20 min after the temperature shift. (**D**) Data form (**B**) is plotted to show the ratio of average cohesin levels 60 kb on either side of all 16 centromeres before and 20 min after the temperature shift. (**E and F**) *wapl∆* (K20891) and *scc2-3XmAID wapl∆* (K26831) were arrested in either late G1 or G2 and treated with auxin (IAA) for 60 min (to degrade Scc2) and subjected to Cal-ChIP-Seq. Samples drawn before (0 min) and after (60 min) auxin addition were analysed by calibrated ChIP-sequencing (Scc1-PK6). Cohesin ChIP profiles along chromosomes four is shown. Also see *Figure 1—figure supplement 1H and I*. (**G and H**) Data form (E and F respectively) are plotted to show the ratio of average cohesin levels 60 kb on either side of all 16 centromeres before and 60 min after auxin addition (**I and J**) The Wapl-independent activity that releases cohesin from DNA is active only in G1 (**I**) and not in mitotic cells (**J**).

DOI: https://doi.org/10.7554/eLife.44736.004

Wapl-independent release in the absence of Scc4 activity. Remarkably, calibrated ChIP-seq showed that the temperature shift profiles of *wpl1∆ SCC4* cells and *wpl1∆ scc4-4* cells arrested in late G1 by Sic1 are very similar if not identical (*Figure 3A*). Thus, in contrast to Scc2, inactivation of Scc4 in late G1 has little or no effect on cohesin's chromosomal association. Scc2 is therefore capable of blocking release even when Scc4 is inactive.

Our finding that Scc4 is required for de novo loading (*Figure 3—figure supplement 1D*) (*Petela et al., 2018*) but not for maintaining cohesin's chromosomal association (*Figure 3A*) confirms that there is indeed little or no turnover of chromosomal cohesin in late G1 cells lacking Wapl, which is consistent with imaging studies both in yeast (*Chan et al., 2012*) and mammalian cells (*Rhodes et al., 2017a*). Scc2 therefore maintains chromosomal cohesin because it actively hinders release not because it merely re-loads cohesin that has been released (*Figure 3B*).

## Wapl-independent RA does not require Pds5

Pds5 is required for Wapl-mediated release and could in principle have a more fundamental role in the release mechanism than Wapl itself. The finding that Scc2 transiently displaces Pds5 from cohesin during the act of loading at centromeres (*Petela et al., 2018*) suggests that Scc2 might block release by displacing Pds5. If so, depletion of Pds5 from late G1 cells using an auxin-dependent degron should abrogate cohesin's dissociation from the genome induced by inactivation of Scc2. To test this, we pre-synchronised *PDS5-AID* and *PDS5-AID scc2-45* in early G1 using α factor before releasing them in the presence of auxin into a Sic1-mediated late G1 arrest (*Figure 3—figure supplement 1B*). Calibrated ChIP-seq revealed a major difference in the temperature shift ratio profiles of the two strains (*Figure 3C* and *Figure 3—figure supplement 1C*). Notably, in the absence of Pds5, the temperature shift increased association in *SCC2* (blue) but decreased it in *scc2-45* (red) cells. The significant gap between blue and red curves implies that Scc2 is required to maintain cohesin's association with chromosomes even in the absence of Pds5 (*Figure 3C* and *Figure 3—figure supplement 1C*).

## Wapl-independent RA involves Smc ATPases

A key question is whether Wapl-dependent and -independent release share the same basic mechanism. A property of Wapl-dependent release is its abrogation by mutation of highly conserved residues in Smc1 and Smc3's ATPases, namely the signature motif *smc1L1129V*, D-loop *smc1D1164E*, and H-loop *smc3T1185M* mutations, which all restore viability to *eco1∆* cells. (*Çamdere et al., 2015*; *Elbatsh et al., 2016*; *Huber et al., 2016*). This raises the possibility that cohesin's ATPase has an important role in Wapl-dependent release. To address whether this feature is shared by Wapl-independent release, we measured the effect of inactivating *scc2-45* in *smc1D1164E scc2-45* and *smc3T1185M scc2-45* double mutants. Calibrated ChIP-seq revealed that inactivation of Scc2 had little effect on chromosomal *smc1D1164E* cohesin (*Figure 4A*) and only a modest effect on *smc3T1185M* (*Figure 4—figure supplement 1A and B*). These data imply that Wapl-independent release induced by Scc2 inactivation occurs by the same mechanism as release mediated by Wapl in the presence of Scc2.

The abrogation of release by *smc1D1164E* reveals an interesting conundrum. Scc2 activates cohesin's ATPase and might therefore block release by de-stabilizing the engagement of Smc1/3 ATPase heads. And yet, a release mechanism unleashed by Scc2's inactivation is eliminated by a *smc1*

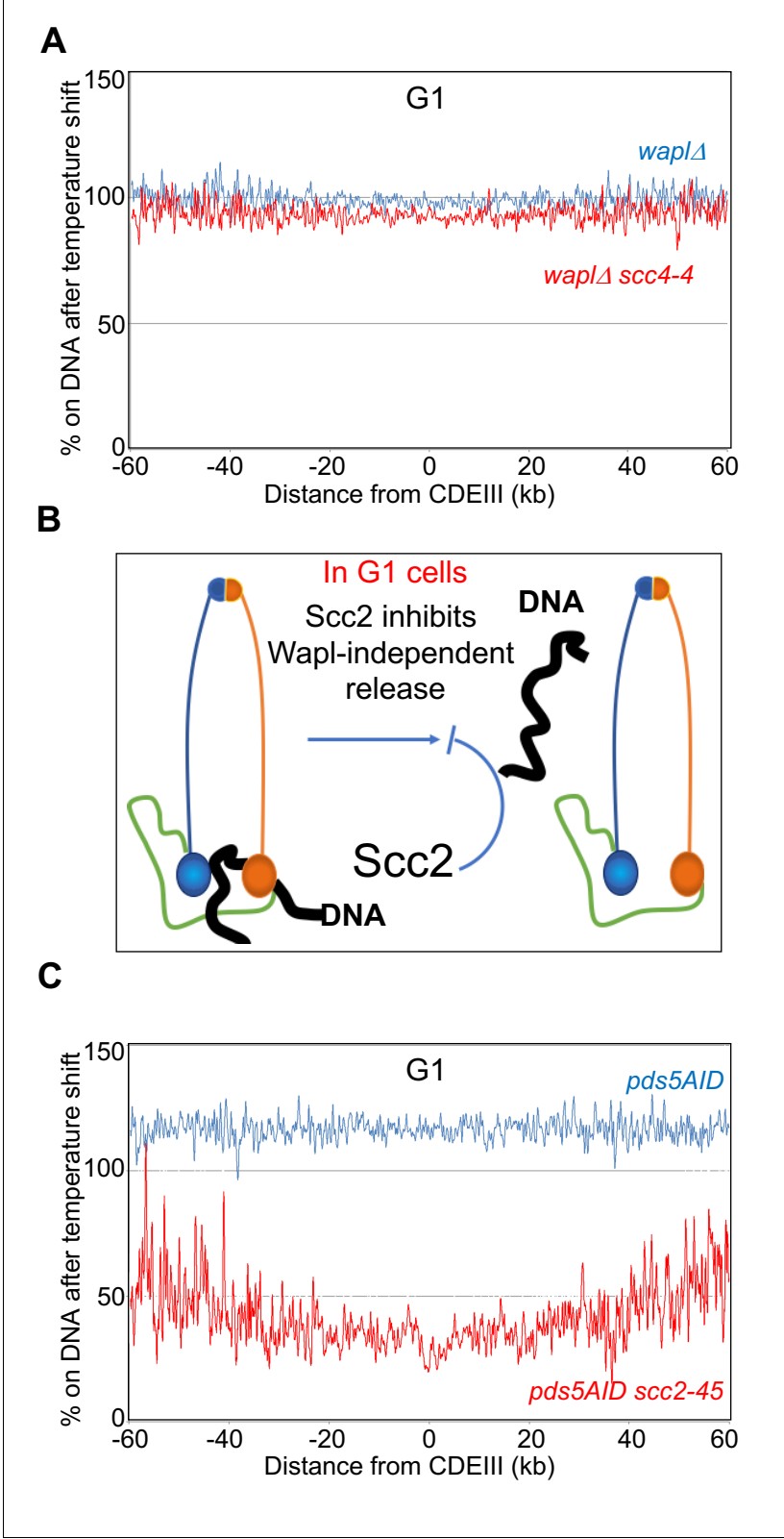

**Figure 3.** Pds5 is not required for the Wapl-independent release and Scc2 does not require Scc4 to inhibit the Wapl-independent release. (**A**) *wapl∆* (K27569) and *wapl∆ scc4-4* (K27570) strains were arrested in late G1 at 25°C and subjected to temperature shift to 37°C for 30 min. Ratio of average cohesin levels before and 30 min after the temperature shift is plotted. Also see S2A and D. (**B**) Association with Scc4 is not required for Scc2 to block the

*Figure 3 continued on next page*

*Figure 3 continued*

Wapl-independent release. Scc2 actively inhibits release in G1. (C) *pds5-AID* (K26415) and *pds5-AID scc2-45* (K26414) strains were arrested in G1 with α-factor and released into sic1(late G1) arrest in the presence of auxin (IAA) and subjected to temperature shift and Cal-ChIP-Seq. Ratio of average cohesin levels before and temperature shift is plotted. Also see *Figure 3—figure supplement 1B and C*.
DOI: https://doi.org/10.7554/eLife.44736.005

The following figure supplement is available for figure 3:

**Figure supplement 1.** Pds5 is not required for the Wapl-independent release and Scc2 does not require Scc4 to inhibit the Wapl-independent release.
DOI: https://doi.org/10.7554/eLife.44736.006

mutation thought to abrogate release by abolishing ATPase activity (*Çamdere et al., 2015*; *Elbatsh et al., 2016*). If *smc1D1164E* really eliminated ATP hydrolysis, then Scc2 cannot prevent release by inducing ATP hydrolysis. We therefore purified wild type (WT), walker B mutants in both Smc1 and Smc3 ATPase (1EQ 3EQ), *smc1D1164E* mutant cohesin tetramers and compared their ATPase activity stimulated by Scc2 in the presence and absence of DNA (*Figure 4—figure supplement 1C*). As expected, the activity of wild type cohesin was fully dependent on Scc2 and stimulated by DNA while the Walker B mutant complex was inactive under all conditions. *smc1D1164E* reduced activity about fourfold, an effect that was largely suppressed by the presence of DNA (*Figure 4—figure supplement 1C*). Thus, suppression of RA by *smc1D1164E* is not necessarily due to an adverse effect on cohesin's ATPase activity. The Smc3 walker B mutant, *smc3E1155Q* (that is capable of ATP binding and not hydrolysis) associates with Scc2 at centromeric loading sites (*Petela et al., 2018*). Our finding that *smc1D1164E* reduced the ability of cohesin containing the *smc3E1155Q* walker B mutant to associate with Scc2 at centromeric loading sites by 50% (*Figure 4—figure supplement 1D*) raises the possibility that *smc1D1164E* also affects a process prior to ATP hydrolysis.

## Wapl-independent RA disengages the Smc3/Scc1 interface

If Wapl-independent release shares a mechanism with Wapl-dependent release, then it should be abrogated by preventing dissociation of the Smc3/Scc1 interface (*Chan et al., 2012*) (*Beckouët et al., 2016*). We therefore tested whether cohesin containing a Smc3-Scc1 fusion protein, which restores viability to *eco1Δ* cells by inactivating Wapl-dependent release, persists on chromosomes upon Scc2 inactivation in late G1 cells. Association with chromosomes of the Smc3-Scc1 fusion protein depends on Scc2 (*Guacci et al., 2019*). Moreover, Scc2 inactivation does not affect the stability of the fusion protein (*Figure 4—figure supplement 1E*). Calibrated ChIP-seq shows that unlike Scc1-PK (*Figure 2*), chromosomal association of a Smc3-Scc1-PK fusion protein is unaffected by Scc2 inactivation (*Figure 4B*). This suggests that Wapl-independent release blocked by Scc2 in G1 cells involves disengagement of the Smc3/Scc1 interface.

To address whether inactivation of Scc2 in G1 *wpl1Δ* cells actually induces dissociation of Scc1's NTD from Smc3, we used a version of Smc3 with a functional cysteine substitution within its coiled coil (S1043C) that can be efficiently crosslinked to a natural cysteine within Scc1's NTD (C56) using the homobifunctional sulfhydryl active reagent Bis-maleimidoethane (BMOE) (*Gligoris et al., 2014*). Though rapidly degraded in wild type cells, the Scc1 (1-181) N-terminal fragment created by separase remains stably associated with Smc3 in *wpl1Δ* mutants (*Beckouët et al., 2016*), a phenomenon readily observed by cross-linking Smc3S1043C to Scc1C56 (*Figure 4C*). Using this assay, we compared Smc3 NScc1 crosslinking in wild type, *scc2-3XmAID*, *wpl1Δ*, and *wpl1Δ scc2-3XmAID* cells upon addition of auxin. Scc2 depletion had little effect in *WPL1* (wild type) cells (*Figure 4D,E* and *Figure 4—figure supplement 1F*), where Wapl promotes release whether or not Scc2 is present. As expected, *wpl1Δ* increased the amount of NScc1 associated with Smc3 (*Figure 4E*) and elevated Smc3-NScc1 crosslinking (*Figure 4D*). Under these circumstances, Scc2 depletion was accompanied by reduced NScc1's association with and cross-linking to Smc3 (*Figure 4D and E*), suggesting that Scc2 is necessary to prevent NScc1's dissociation from Smc3 in *wpl1Δ* cells.

Part of the reduction in Smc3-NScc1 crosslinking in *wpl1Δ* cells induced to degrade Scc2 can be attributed to the reduced activity of the Scc2-3XmAID fusion protein even in the absence of auxin. Importantly, auxin addition reduced the cross-linking in *wpl1Δ scc2-3XmAID* cells but not in *wpl1Δ* controls (*Figure 4F*), demonstrating that acute depletion of scc2-3XmAID (*Figure 4—figure*

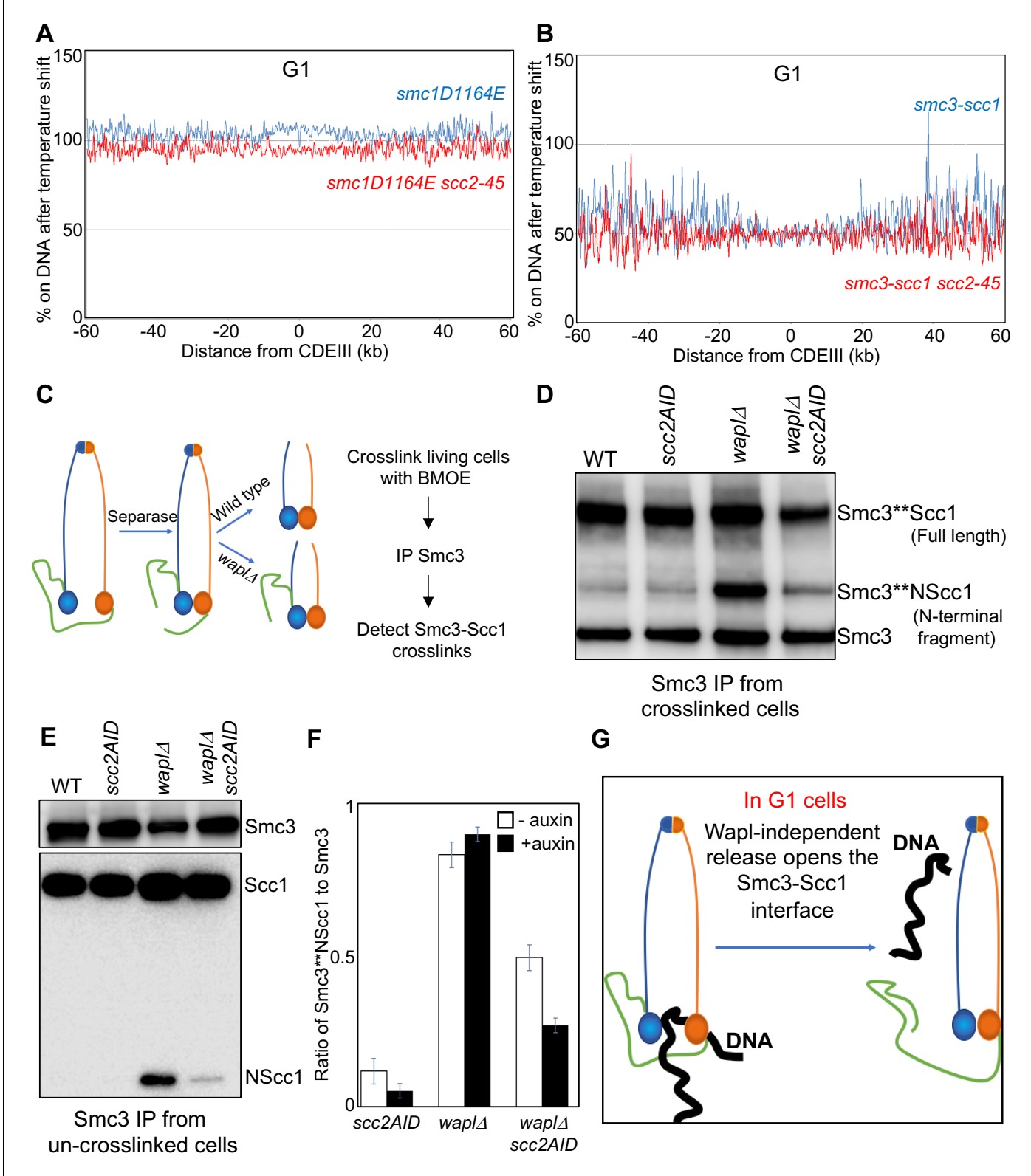

**Figure 4.** Wapl-independent release requires Smc ATPases and involves dissociation of the Smc3-Scc1 interface. (**A**) *smc1D1164E* (K26765) and *smc1D1164E scc2-45* (K26766) strains were arrested in late G1 at 25°C and subjected to temperature shift to 37°C for 20 min. Ratio of average cohesin levels before and 20 min after the temperature shift is plotted. (**B**) Strains expressing a covalently fused Smc3-Scc1 fusion protein, *smc3-scc1* (K26994) and *smc3-scc1 scc2-45* (K26993) were arrested in late G1 at 25°C and subjected to temperature shift to 37°C for 20 min. Ratio of average cohesin levels

*Figure 4 continued on next page*

*Figure 4 continued*

before and 20 min after the temperature shift is plotted. Also see *Figure 4—figure supplement 1E*. (C) Scc1 is cleaved by Separase in Anaphase. The Separase cleaved N-Terminal fragment of Scc1 (NScc1) remains stably associated with Smc3 in cells lacking Wapl mediated releasing activity. This interaction can be measured by crosslinking the two proteins in vivo using BMOE as detailed in Materials and Methods; Cys substitution of a Ser residue at 1043 in Smc3 allows the crosslinking of the 1043C with a natural Cys at position 56 in Scc1. (D) Wild type (K22156), *scc2-3XmAID* (28095), *waplΔ* (K28094) and *waplΔ scc2-3XmAID* (K28096) strains expressing Smc3 (S1043C)-HA3 were arrested in G2 with nocodazole. Subsequently, 5 mM auxin was added to the G2 arrested cultures and incubated for 60 min (See *Figure 4—figure supplement 1F*). This was followed by in vivo crosslinking and Smc3 IP as detailed in Materials and Methods. Smc3-HA3 immunoprecipitated from whole-cell extracts was analysed by western blotting detecting the HA epitope. (E) Smc3-HA3 immunoprecipitated from whole-cell extracts of strains grown (D) that were not subjected to in vivo crosslinking. The IP was analysed by western blotting against HA and MYC epitopes, the bands corresponding to full length Scc1 and the Scc1 N-terminal fragment are marked. (F) *scc2-3XmAID* (28095), *waplΔ* (K28094) and *waplΔ scc2-3XmAID* (K28096) strains expressing Smc3 (S1043C)-HA3 were arrested in G2 with nocodazole. The cultures were either treated with 5 mM auxin or left untreated for 60 min. This was followed by in vivo crosslinking and western blotting to detect Smc3-Scc1 crosslinks. Three independent repetitions of the experiment were quantified using the LI-COR odyssey software to measure the intensities of Smc3 and Smc3-NScc1. The ratio of Smc3-NScc1 band intensity to that of the Smc3 band is plotted. (G) Wapl-independent releasing activity causes disengagement of the Smc3-Scc1 interface.

DOI: https://doi.org/10.7554/eLife.44736.007

The following figure supplement is available for figure 4:

**Figure supplement 1.** Wapl-independent release requires Smc ATPases.

DOI: https://doi.org/10.7554/eLife.44736.008

*supplement 1F*) does indeed trigger a reduction of Smc3-NScc1 crosslinking. Together with the persistence on chromosomes of cohesin containing the Smc3-Scc1 fusion protein (*Figure 4B*), these results suggest that Wapl-independent release induced by Scc2 inactivation in G1 cells is mediated by opening the Smc3/Scc1 (*Figure 4G*). Wapl-independent and -dependent release clearly use the same fundamental mechanism.

## Neither Smc3 acetylation nor Pds5 are required to turn off Wapl-independent release

How is Wapl independent release turned off when cells enter G2? Because it shares a similar mechanism to the Wapl-dependent process, Smc3 acetylation could be responsible (*Beckouët et al., 2016*; *Rowland et al., 2009*). To test this, we arrested *ECO1 scc3K404E scc2-45* and *eco1Δ scc3K404E scc2-45* cells in G2/M at 25°C and then shifted both to 37°C for 20 min. Surprisingly, inactivation of Scc2 had no effect on cohesin's association with the genome in either culture (*Figure 5A and B*). This was confirmed using sedimentation velocity/gel electrophoresis to measure sister minichromosome cohesion, which was unaffected by Scc2 inactivation both in the presence and absence of Eco1 (*Figure 5—figure supplement 1A*). The transition from a state in which Scc2 is required to block Wapl -independent release (G1) to one where it is not (G2) does not therefore require cohesin's acetylation by Eco1.

Because Pds5 is necessary to maintain cohesion, even in the absence of Wapl, it is conceivable that changes in Pds5's behaviour might be involved. To test this, we synchronised *PDS5-AID* or *PDS5-AID scc2-45* cells in G1 and then allowed them to undergo replication in the presence of IAA and nocodazole (to arrest cells in G2/M). Calibrated ChIP-seq showed that shifting cells to 37°C for 20 min reduced cohesin's chromosomal association by 50% (*Figure 5—figure supplement 1B–D*), but importantly there was little difference between *SCC2* (blue) and *scc2-45* (red) cells. Thus, Pds5 is unnecessary for shutting off Wapl-independent release when cells enter G2/M (compare *Figure 2A* and *Figure 5—figure supplement 1B*). Note that Pds5 is required for Smc3 acetylation and the fact that Pds5 depletion does not prevent inactivation of Wapl-independent release upon entry into G2/M confirms that acetylation is unnecessary.

## Neither cohesion establishment nor passage through S phase are required to turn off Wapl-independent release

To address whether establishment of cohesion or passage through S phase is required for the switch, we asked whether Scc2 is required to prevent release of cohesin loaded onto chromosomes only during G2. To this end, *scc3K404E* and *scc3K404E scc2-45* cells containing an ectopic copy of a PK tagged *SCC1* gene under control of the *GAL* promoter were arrested in G2/M at 25°C and a

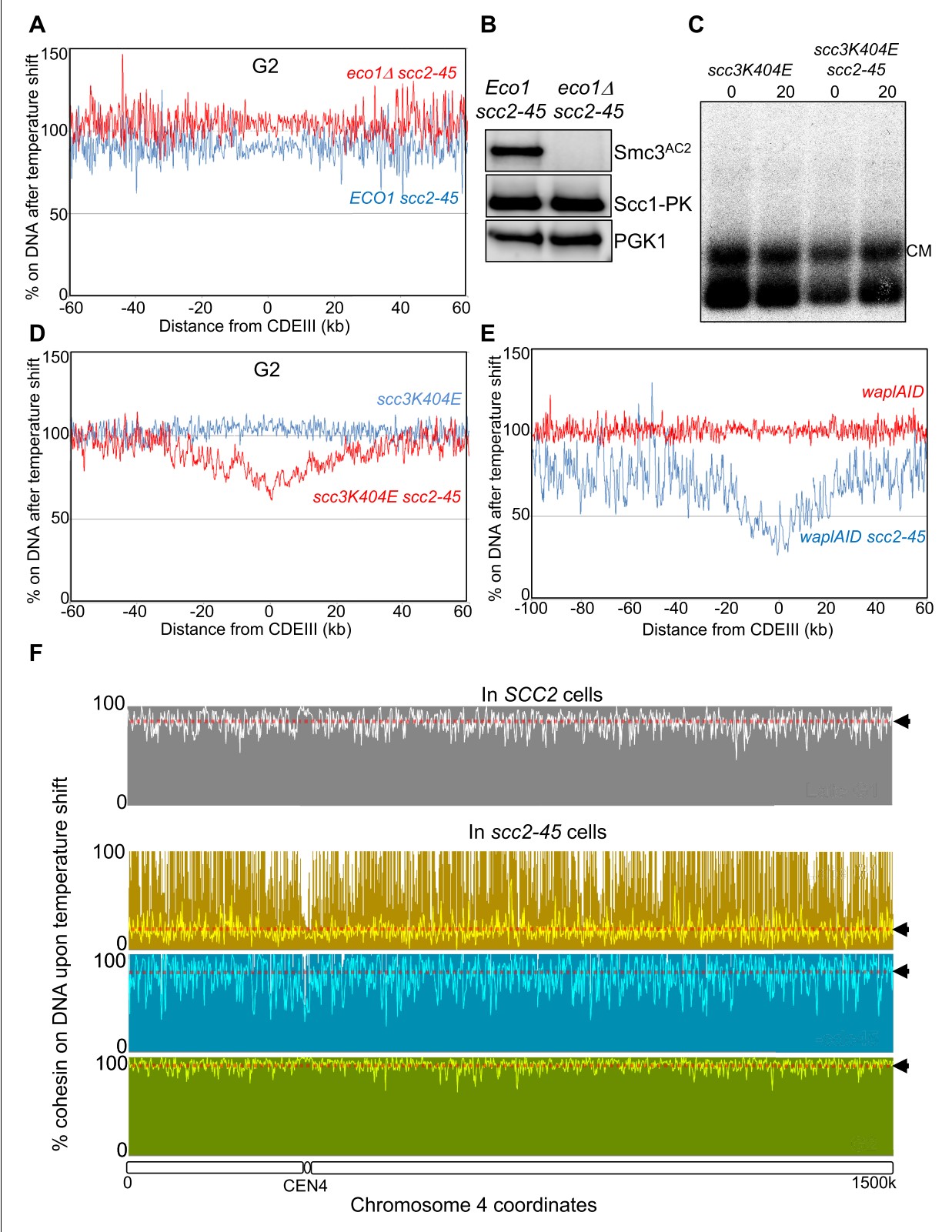

**Figure 5.** Smc3 acetylation and DNA replication are not required to abolish Wapl-independent release. (**A**) *scc3K404E scc2-45* (K24709) and *smc3K404 eco1Δ scc2-45* (K25947) strains arrested in G2 at 25°C and subjected to temperature shift to 37°C for 20 min. Ratio of average cohesin levels before and 20 min after the temperature shift is plotted. (**B**) Strains grown in (**A**) were analysed by western blotting against the indicated antibodies. (**C**) *scc3K404E* (K24697) and *scc3K404E scc2-45* (K24738) strains containing 2C Smc1, 2C Smc3 and galactose inducible 2C Scc1[NC] were arrested in G2 in YEP raffinose

*Figure 5 continued on next page*

*Figure 5 continued*

and Scc1$^{NC}$ expression induced by addition of galactose. 60 min after galactose addition, glucose was added to the cultures and temperature shifted to 37°C. 0- and 20 min samples were subjected to mini-chromosomeIP (Scc1-PK6) following in vivo cohesin crosslinking with BMOE. (D) 0- and 20 min samples from (C) were subjected to Cal-ChIP-Seq. ratio of average cohesin levels before and temperature shift is plotted. Also see *Figure 5—figure supplement 2B*. (E) Cdc45 was depleted from *wapl−AID cdc45-AID* (K27169) and *wapl-AID cdc45-AID scc2-45* (K27168) strains like described in Materials and Methods. Following temperature shift, 0- and 20 min samples of the cdc45 depleted strains were subjected to Cal-ChIP-Seq. ratio of average cohesin levels before and temperature shift from −100 kb to +60 KB relative to all 16 centromeres is plotted. Also see *Figure 5—figure supplement 2C and D*. (F) The percentage of cohesin that remains on DNA upon Scc2 inactivation (after temperature shift) along the entire chromosome four is shown. The median cohesin level along the entire chromosome 4 (dotted line) is marked with arrowheads. Data from SCC2 wild type cells arrested in lateG1 (*Figure 2A*) is shown in grey. Data from *scc2-45* cells arrested in late G1 (*Figure 2A*) is shown in yellow. Data from Cdc45 depleted *scc2-45* cells (*Figure 5E*) is shown in turquoise. Data from *scc2-45* cells arrested in G2 (*Figure 2B*) is shown in green.

DOI: https://doi.org/10.7554/eLife.44736.009

The following figure supplements are available for figure 5:

**Figure supplement 1.** Neither Smc3 acetylation nor Pds5 are required to turn off Wapl-independent release.

DOI: https://doi.org/10.7554/eLife.44736.010

**Figure supplement 2.** DNA replication is not required to turn off Wapl-independent release.

DOI: https://doi.org/10.7554/eLife.44736.011

---

pulse of Scc1-PK produced by transient induction with galactose for 60 min. Scc1-PK synthesis was subsequently blocked by transferring cells to glucose, after which the temperature was shifted to 37° C for 20 min. Due to cysteines within *SMC1*, *SMC3*, and *SCC1-PK* alleles, the Scc1-PK tagged cohesin produced by this protocol was 6C, which permitted measurement of mini-chromosome CMs and CDs. This revealed that cohesin which had loaded during G2 in the absence of conventional releasing activity (due to *scc3K404E*) formed CMs but few if any CDs (*Figure 5C*).

As expected, the de novo formation of CMs during G2 depended on Scc2 (*Figure 5—figure supplement 2A*). Crucially, these CMs were largely unaffected by Scc2's inactivation in *scc2-45* cells (*Figure 5C*), implying that cohesin loaded onto chromosomes during G2 without forming cohesion does not depend on Scc2 for its maintenance and is therefore not subject to Wapl-independent release. Calibrated ChIP-seq revealed that cohesin along chromosome arms was similarly unaffected by Scc2's inactivation while peri-centric cohesin was only modestly reduced (*Figure 5D* and *Figure 5—figure supplement 2B*). Thus, cohesin's switch to a stable form (resistant to Scc2 inactivation) does not require prior association with chromosomes during S phase or indeed establishment of cohesion.

## DNA replication is not required to switch off Wapl-independent release along chromosome arms

To address whether DNA replication is required for the switch, we analysed the consequences of depleting Cdc45, an essential component of the CMG helicase. *wapl-AID CDC45-AID* and *wapl-AID scc2-45 CDC45-AID* cells were released from G1 arrest in the presence of synthetic auxin (NAA) and nocadozole. Though Cdc45-depleted cells fail to replicate DNA (*Figure 5—figure supplement 2C*) or acetylate Smc3 (*Figure 5—figure supplement 2D*), they nevertheless degrade Sic1 and accumulate modest levels of the mitotic Clb2 cyclin, albeit less than cells allowed to replicate (*Figure 5—figure supplement 2D*). Calibrated ChIP-seq revealed that Scc2 inactivation caused a two-fold drop in peri-centric chromosomal cohesin but only a modest change in chromosome arm cohesin (*Figure 5E*). Importantly, the effect of inactivating Scc2 in Cdc45-depleted cells more closely resembles the G2 pattern than the G1 pattern (*Figure 5F*), at least along chromosome arms, which represents the vast majority of chromosomal cohesin (in the case of chromosome IV, peri-centric cohesin accounts for less than 10%). These data are consistent with the notion that it is activation of Cdk1 rather than replication per se that switches off cohesin's release from chromosomes upon Scc2 inactivation.

## Cdk1 is not required for cohesin to persist on chromosomes without Scc2

If Cdk1 is responsible for switching off release when cells initiate S phase, then its inhibition should cause CMs made in G2 to revert to a state that requires Scc2 for their maintenance. To test this, we

used the analogue sensitive *cdc28-as1* allele that can be inhibited in a highly specific manner by addition of the ATP analogue 1NMPP1 (*Bishop et al., 2000*). *cdc28-as1* cells were arrested in G2/M with nocodazole, whereupon Cdk1 was inhibited by addition of 1NMPP1. As expected, this caused Clb2 degradation and Sic1 accumulation, which ensures complete inhibition of Clb/Cdk1 kinases (*Figure 6A*). However, as previously observed (*Amon, 1997*), these events were accompanied by rapid degradation of Scc1 (*Figure 6A*), presumably due to separase activation. To prevent this, we used the *GAL* promoter to express Scc1^NC-PK, a PK-tagged allele (*scc1R180D R268D*) that cannot be cleaved by separase (*Uhlmann et al., 1999*). Importantly, the cohesin generated by Scc1^NC-PK was 6C. *scc3K404E cdc28-as1* and *scc3K404E scc2-45 cdc28-as1* cells were arrested in G2/M, Scc1^NC-PK was expressed for 60 min by addition of galactose. After shutting off further Scc1^NC-PK synthesis, Cdk1 was inhibited by addition of 1NMPP1, and 60 min. later cells were shifted from 25°C to 37°C to inactivate Scc2. Remarkably, the CMs produced prior to Cdk1's inhibition were unaffected by Scc2's inactivation (*Figure 6B*). Cdk1 activity is therefore not required to maintain chromosomal cohesin in a state resistant to Scc2 inhibition.

The previous experiment shows that cohesin loaded onto chromosomes in G2 is refractory to Scc2 inactivation even when Clb/Cdk1 is inhibited and yet cohesin loaded onto chromosomes in cells arrested in late G1 with inactive Clb/Cdks is not (*Figures 1* and *2*). If Cdk1 activity were really the parameter that determines whether cohesin requires the Scc2 maintenance function, then the cohesin in these two populations should have behaved identically. One possible explanation for their different behaviour is that the low Cdk1 state created by inhibition of Cdk1 in G2 cells is in some way different from the low Cdk1 state created by expression of non-degradable Sic1. In other words, these two cell cycle states in fact differ in some unknown way that affects cohesin's behaviour. If this is the case, even cohesin loaded onto chromosomes after Cdk1 had been inhibited in G2 cells would behave differently to that loaded in Sic1-arrested cells.

To address if cohesin loaded onto chromosomes in the presence of the low Cdk1 levels produced by treating G/M phase arrested *cdc28-as1* cells with 1NMPP1 is sensitive or resistant to Scc2 inhibition, we arrested *scc3K404E SCC2 cdc28-as1* and *scc3K404E scc2-45 cdc28-as1* cells in G2/M, inhibited Cdk1 by 1NMPP1, and then only subsequently induced Scc1^NC-PK to generate CMs. Crucially, these CMs, unlike those made prior to Cdk1 inhibition (*Figure 6B*), disappeared in *scc2-45* but not in the *SCC2* cells upon shift from 25°C to 37°C (*Figure 6C*). This implies that the G1-like state created by inhibition of Cdk1 in G2/M cells is in fact similar to that produced by arresting cells in late G1 with non-degradable Sic1. CMs created in both types of G1 states require Scc2 for their maintenance.

This suggests that the persistence of a G2 character following Cdk1 inhibition of CMs made in G2 cells, namely their resistance to Scc2 inhibition, is conferred by cohesin's continued association with chromosomes. In other words, loss of Scc2 resistance characteristic of G2 chromosomal cohesin, upon Cdk1 inhibition, requires cohesin's removal from chromosomes. If the resistance is conferred by a post-translational modification promoted by Cdk1, then the modification must persist even when Cdk1 is subsequently inhibited, as long as cohesin remains associated with chromosomes. Though this might seem unlikely, there is in fact a clear precedent for such behaviour, namely the dependence of Smc3 deacetylation by Hos1 on Scc1 cleavage and not the decline in Cdk1 activity that normally accompanies cleavage during anaphase (*Beckouët et al., 2010*).

## In the absence of all forms of cohesin release, Scc2 becomes dispensable for cohesion establishment

Our discovery that only CMs are produced when Scc1 is induced in G2/M phase cells and that these CMs are refractory to Scc2 inactivation even when Cdk1 is inhibited provided a means of testing whether Scc2 is required to convert CMs to CDs. Inhibition of Cdk1 in G2/M, by inducing high levels of Sic1, induces formation of pre-RCs, from which a new round of replication can be triggered when Cdk1 is re-activated (*Dahmann et al., 1995*). To establish such conditions using 1NMPP1, we arrested *cdc28-as1* cells in G2/M and inhibited Cdk1 with 1NMPP1 for 60 min. Because Cdk1 inhibition under these conditions is not accompanied by cytokinesis, cells retain their 2C DNA content. The culture was then filtered and split, with one half incubated for a further 90 min in medium containing DMSO (Cdk1 re-activation) and the other in medium containing both DMSO and 1NMPP1 (continued Cdk1 inhibition) (*Figure 7A*). Western blotting and FACs analysis showed that Cdk1 re-activation (but not continued inhibition) was accompanied by Sic1 degradation, Clb2 accumulation,

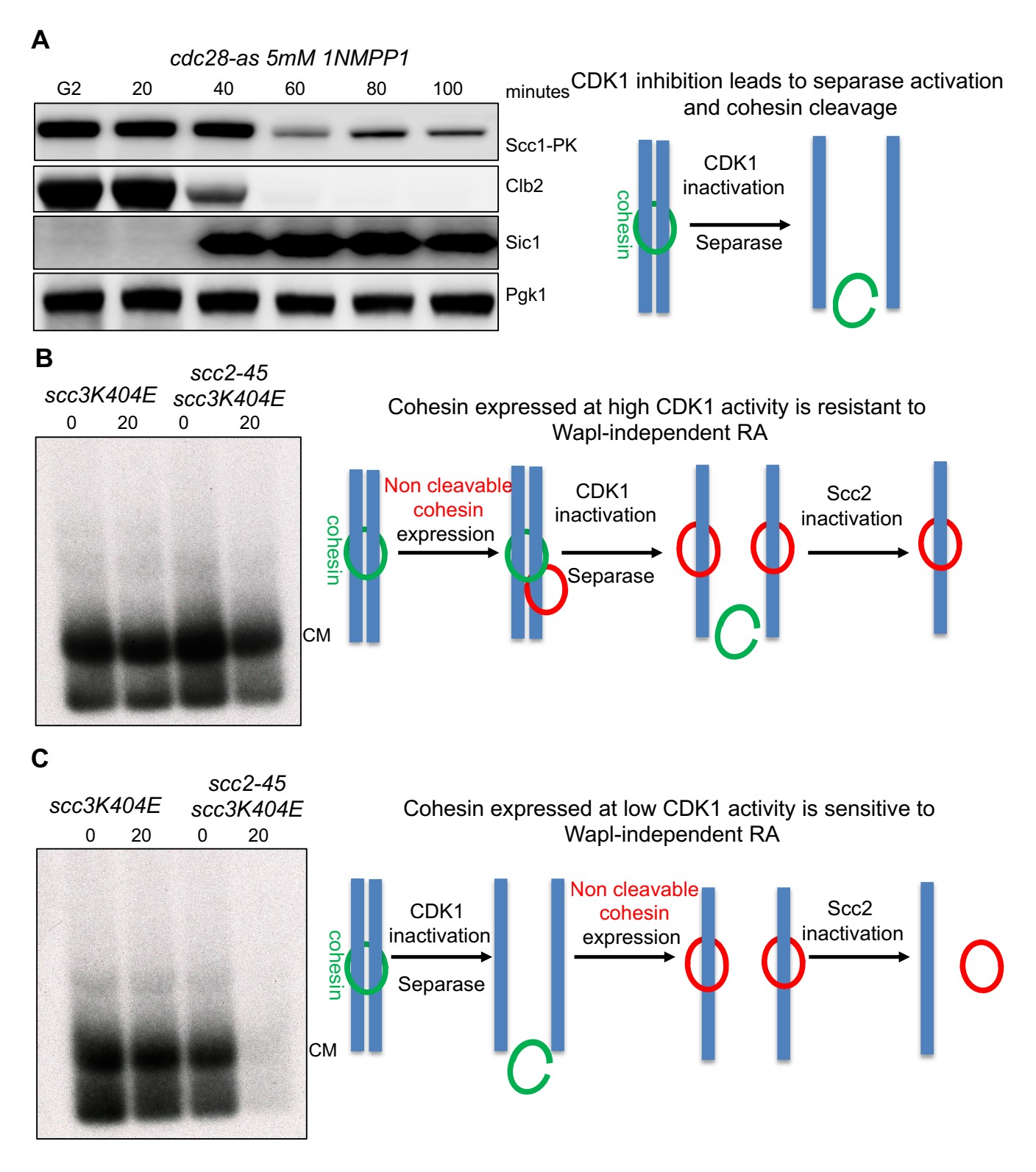

**Figure 6.** Cohesin expressed at high CDK1 levels is resistant to Wapl -independent release. (**A**) CDK1 inhibition leads to cohesin cleavage: *cdc28-as1* (K25423) cells were arrested in G2 with nocadazole and treated with 5 µM 1NMPP1. Samples were drawn at indicated times and subjected to western blot analysis with the indicated antibodies. (**B**) Cohesin expressed at high CDK1 levels is resistant to Wapl-independent release: *scc3K404E cdc28-as1* (K25437) and *scc3K404E scc2-45 cdc28-as1* (K25440) containing 2C Smc1 and 2C Smc3 along with galactose inducible 2C Scc1[NC] were arrested in G2
*Figure 6 continued on next page*

*Figure 6 continued*

with nocadazole. Scc1$^{NC}$ expression induced by galactose addition for 60 min. Glucose and 1NMPP1 were added to the cultures for 60 min followed by temperature shift to 37°C for 20 min. Samples drawn before (0) and after temperature shift (20) were analysed by mini-chromosome IP following in vivo cohesin crosslinking with BMOE. (C) CDK1 is required to abolish Wapl-independent release: Strains described in (B) were arrested in G2. Followed by 1NMPP1 addition for 60 min. After this, galactose was added to the cultures to induce 2C Scc1$^{NC}$ for 60 min. Glucose was added to the cultures followed by temperature shift to 37°C for 20 min. Samples drawn before (0) and after temperature shift (20) were analysed by mini-chromosome IP following in vivo cohesin crosslinking with BMOE.

DOI: https://doi.org/10.7554/eLife.44736.012

and as expected the appearance of cells with a 4C DNA content due to re-replication (*Figure 7B and C*). Crucially, when this protocol was used to induce re-replication at 37°C (*Figure 7—figure supplement 1B*), conversion to CDs of CMs created by Scc1$^{NC}$-PK during G2/M (*Figure 7D*) was similar if not identical in *SCC2 scc3K404E cdc28-as1* and *scc2-45 scc3K404E cdc28-as1* cells (*Figure 7E*).

Because *scc2-45* cells are incapable of forming either CMs or CDs at 37°C when allowed to replicate after release from a G1 arrest (*Figure 7—figure supplement 1A*) (*Srinivasan et al., 2018*), the CDs that appear in *scc2-45* cells when re-replication is induced by transient Cdk1 inhibition (*Figure 7E*) were presumably derived from the Scc2-resistant CMs produced during the previous G2 arrest. In other words, CMs can be converted during S phase to CDs in the absence of Scc2 activity. Though unexpected, the result was highly reproducible (*Figure 7—figure supplement 1C*). Importantly, the CDs formed in the absence of Scc2 activity was entirely dependent on cohesin crosslinking with BMOE (*Figure 7—figure supplement 1D*), confirming that the CDs formed in the absence of Scc2 were held together by cohesin and not by DNA catenation.

Because of the unexpected results from this experiment and because it merely addressed CD formation by small circular mini-chromosomes, we sought an alternative way of addressing Scc2's role during S phase. Our goal was to measure establishment of cohesion along chromosome arms in cells that enter S phase under very different circumstances. Because Scc2 is unnecessary to maintain cohesin's association with chromosomes in Wapl deficient cells that activate Cdk1 but cannot undergo replication due to Cdc45 depletion (*Figure 5*), we reasoned that the same might be true for cells whose replication is prevented by hydroxyurea (HU). We therefore asked whether or not Scc2 is required to generate cohesion upon replication following a transient HU arrest.

To measure cohesion on a chromosome arm, we used a version of chromosome V in which multiple tandem TetO arrays at the *URA3* locus are marked by TetR-GFP (*Michaelis et al., 1997*). *SCC2* or *scc2-4* cells were released from G1 arrest into HU containing medium at 25°C. After 45 min, cells were transferred to HU-free medium at 35.5°C under conditions in which cells were depleted for Cdc20, which prevented separase activation and caused metaphase arrest. Crucially, calibrated whole genome sequencing revealed little or no origin firing during the 45 min incubation in the presence of HU (*Figure 7—figure supplement 1E*).

To assess the ability of these cells (*Figure 7F* red graphs) to build cohesion, they were compared to a different population of cells that were released from G1 arrest into HU-free medium directly at 35.5°C (*Figure 7F* blue graphs). In *SCC2* cells, the fraction of cells with two GFP dots (a measure of cohesion loss) remained low throughout the time course whether or not they had been given an opportunity to load cohesin in the presence of HU at 25°C (see *SCC2 pds5S81R* cells in left panel *Figure 7F*). Thus, as expected, wild type cells established cohesion under both regimes.

In *scc2-4* cells, the fraction of cells with two GFP dots rose soon after replication, indicating a failure to establish sister chromatid cohesion. Crucially, the cohesion defect was only modestly reduced when cells were allowed to load cohesin onto chromosomes in the presence of HU at 25°C (compare blue and red curves in middle panel *Figure 7G*). This implies that loading of cohesin onto chromosomes during the HU arrest at 25°C is insufficient to create efficient cohesion when cells are released from the HU arrest at 35.5°C. Due to Cdk1 activation, Wapl-independent release should be inactive in HU arrested cells but Wapl-dependent release should still take place, as very little Smc3 acetylation occurs during the 45 min HU arrest (*Nasmyth, 2017*) and because Wapl can induce release in *eco1* mutants arrested in G2/M (*Chan et al., 2012*; *Srinivasan et al., 2018*).

To test whether Wapl-dependent release contributes to the lack of cohesion establishment, we introduced the *pds5S81R* mutation, which abolishes Wapl-dependent release (*Rowland et al.,*

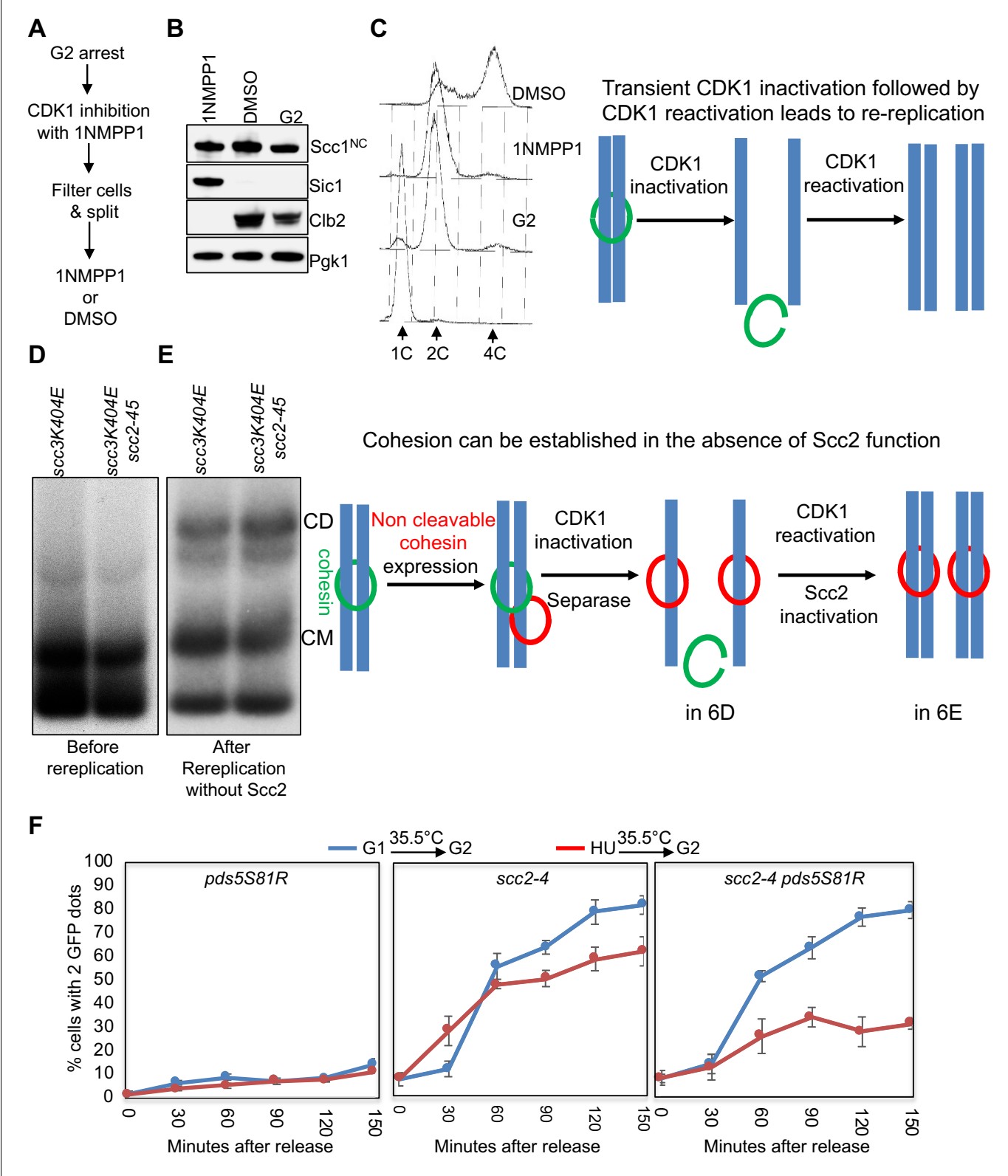

**Figure 7.** In the absence of all forms of cohesin release, Scc2 becomes dispensable for cohesion establishment. (A–C) Transient inhibition of CDK1 in mitotic cells induces re-replication: (A) Growth conditions to induce re-replication: *scc3K404E cdc28-as1* (K25437) containing 2C Smc1 and 2C Smc3 along with galactose inducible 2C Scc1NC was arrested in G2 with nocadazole. Scc1NC expression induced by galactose addition for 60 min. After this, glucose and 1NMPP1 were added to the cultures. After 60 min, the culture was shifted to 37°C for 20 min. Subsequently, the culture was filtered, *Figure 7 continued on next page*

*Figure 7 continued*

washed with YEP medium and resuspended into YEPD medium containing either DMSO or 1NMPP1at 37°C. 90 min later, the culture was analysed by western blotting with indicated antibodies (B) and FACS (C). (D and E) in the absence of release, Scc2 is dispensable for cohesion establishment: re-replication was induced in *scc3K404E cdc28-as1* (K25437) and *scc3K404E scc2-45 cdc28-as1* (K25440) expressing galactose inducible 6C non-cleavable cohesin using the growth regime described in (A). Samples drawn before (D) and after (E) induction of re-replication were analysed by in vivo crosslinking and minichromosome IP. (F) Inactivation of releasing activity suppresses the cohesion defect caused by Scc2 inactivation: Sister chromatid cohesion was measured as described in Materials and Methods in *pds5S81R* (K27443) *scc2-4* (K15028) and *scc2-4 pds5S81R* (K27575) strains that were arrested in G1 or S (HU) and released into G2 arrest (by Cdc20 depletion) at non-permissive temperature (35.5°C).

DOI: https://doi.org/10.7554/eLife.44736.013

The following figure supplement is available for figure 7:

**Figure supplement 1.** In the absence of all forms of cohesin release, Scc2 becomes dispensable for cohesion establishment.

DOI: https://doi.org/10.7554/eLife.44736.014

---

*2009*). Remarkably, transient incubation in HU at 25°C in *scc2-4 pdss5S81R* cells greatly ameliorated their cohesion defects following replication at 35.5°C (compare red and blue curves in *Figure 7F* right panel). Cohesion was not however restored to the level in *scc2-4 pds5S81R* cells grown only at permissive temperature (*Figure 7—figure supplement 1F*). We conclude that when both Wapl-dependent and -independent release mechanisms are inactivated, chromosomal cohesin associated with unreplicated genomes can generate substantial sister chromatid cohesion during S phase in the absence of any further Scc2 activity. This result is consistent with our finding that Scc2 is also not required to convert CMs to CDs. (*Figure 7D*).

## Discussion

The work described here requires a major re-appraisal of the roles of Wapl and Scc2 in determining the dynamics of cohesin's association with chromosomes in yeast. First and foremost, Scc2 is not merely a factor that loads cohesin onto chromosomes. In G1 cells lacking Wapl, where chromosomal cohesin does not in fact turnover (*Chan et al., 2012*), release is nevertheless possible but actively suppressed by Scc2. Because it occurs in the absence of Wapl, we refer to cohesin's dissociation upon Scc2 inactivation during G1 as Wapl-independent release. These findings demonstrate that Scc2 is not merely involved in loading cohesin onto chromosomes but acts long afterwards to prevent release. Previous work with mammalian cells showed that Scc2/Nipbl associates transiently but continuously with chromosomal cohesin (*Rhodes et al., 2017a*). Our work suggests that this association is functionally important.

Our second key insight is that in G1 cells Wapl merely facilitates an activity associated with the ATPases of tripartite rings that is inhibited by Scc2. In other words, contrary to prevailing opinion, Wapl is not an intrinsic feature of the release mechanism. Releasing activity during G1, that is actively inhibited by Scc2, is a property associated with tripartite Smc1/Smc3/Scc1 rings bound possibly only by Scc3. Both Wapl-dependent and -independent release reactions are accompanied by dissociation of Scc1's NTD from Smc3's coiled coil and blocked by fusing Smc3 to Scc1. They are also both abrogated by *smc1D1164E*, which alters how the Smc1 and Smc3 ATPase heads interact. They appear therefore to share a common mechanism, involving transient dissociation of Scc1's NTD from Smc3's coiled coil via a process involving cohesin's Smc1/3 ATPase. One problem in this regard is the suggestion that *smc1D1164E* abolishes release by abolishing cohesin's ATPase activity (*Çamdere et al., 2015*; *Elbatsh et al., 2016*). If correct and Scc2 did likewise, then one would have to postulate that Scc2 acts as an inhibitor as well as an activator of cohesin's ATPase (*Petela et al., 2018*), which is difficult to envisage. Our finding that *smc1D1164E* does not in fact abolish cohesin's ATPase activity raises the possibility that the mutation in fact abrogates release by some other mechanism, which avoids the above conundrum. How the dissociation of NScc1 from Smc3 is regulated by cohesin's ATPase heads, how Scc2 blocks this process, and how Wapl somehow overcomes this inhibition are key questions.

The third insight is the discovery that cohesin's dynamics are subject to a novel type of cell cycle control. Wapl-independent release is specific to G1 cells and is inactivated as cells undergo S phase, not by Smc3 acetylation but by Clb/Cdk1 activity. Because this regulation does not involve Pds5, Scc2, or Wapl, we suggest that it is mediated by phosphorylation either of Smc-kleisin rings or Scc3.

We propose that phosphorylation of cohesin by Cdk1 or another mitotic kinase whose activity depends on Cdk1 as well as acetylation by Eco1 prevent exit gate opening during G2/M. We note that there are similarities between this phenomenon and the recent finding that in the absence of Eso1 (Eco1), Wapl induces loss of cohesion in G2 *S.pombe* cells by a pathway involving de-phosphorylation of Rad21 (Scc1) (*Birot et al., 2017*). Thus, in both organisms cohesin phosphorylation may be accompanied by a change in its dynamics. Though similar in this regard, a key difference is that the release regulated by phosphorylation is Wapl-independent in *S.cerevisiae* but Wapl-dependent in *S.pombe*.

The fourth insight stems from the very different behaviour of newly synthesised and pre-existing chromosomal cohesin in G2 cells that have been converted to a G1-like state through Cdk1 inhibition. While the former requires Scc2 to remain on chromosomes the latter does not. In other words, chromosomal cohesin retains its 'G2' behaviour even when a cell's cell cycle regulatory network is switched to a low Cdk1, G1-like state. This persistence is analogous to abnormal retention of Smc3 acetylation caused by a failure to cleave Scc1.

The fifth and last key insight is our discovery that once cohesin has loaded onto chromosomes in the absence of both Wapl-dependent and -independent release, Scc2 becomes dispensable for establishing sister chromatid cohesion. There are two types of explanation for this surprising finding. Either replication forks pass through cohesin rings or they open them up in a manner that either does not lead to dissociation or that is associated with an extremely rapid Scc2 independent re-association, albeit with both sister DNAs. The on rate for cohesin loading onto chromosomes de novo is approximately 33 min (*Hansen et al., 2017*). If the kinetics of re-association at forks obeyed the same rules, then it would be impossible for displaced cohesin to re-associate before it had diffused far away. Thus, if displacement at forks does take place, the kinetics of re-association must be very different to those that normally apply to nucleoplasmic cohesin.

Though our findings do not exclude the possibility that a Scc2 dependent pathway involving entrapment of single stranded DNA associated with lagging strands (*Murayama et al., 2018*) co-exists with the Scc2 independent one, they are inconsistent with the claim that the former is essential. On the other hand, the finding that cohesin loading in mammalian cells is dependent on Mcm2-7 during S phase but not during telophase (*Zheng et al., 2018*) may be pertinent to our finding that entrapment of leading or lagging strands during DNA replication differs from the process of DNA entrapment at other stages of the yeast cell cycle.

# Materials and methods

## Key resources table

| Reagent type (Species) or resource | Designation | Source Or reference | Identifiers | Additional information |
|---|---|---|---|---|
| Strain, strain background (*S. cerevisiae*) | *S. cerevisiae MATa ade2-1 trp1-1 can1-100 leu2-3,112 his3-11,15 ura3 GAL psi+ All following strains are based on this background* | This study | K699 | Materials and methods subsection experimental models |
| Strain, strain background (*S. cerevisiae*) | *S. cerevisiae MATa scc2-4 SCC1-PK9 TETR-GFP::LEU2 TET Os::URA3 trp1:MET 3p-CDC20::TRP1* | This study | K15028 | Materials and methods subsection experimental models |
| Strain, strain background (*S. cerevisiae*) | *S. cerevisiae Mata trp1:smc 3(E1155Q)-PK6::TRP1* | This study | K17409 | Materials and methods subsection experimental models |

*Continued on next page*

*Continued*

| Reagent type (Species) or resource | Designation | Source Or reference | Identifiers | Additional information |
|---|---|---|---|---|
| Strain, strain background (*S. cerevisiae*) | *S. cerevisiae Mata SCC1-PK9::KAN MX rad61Δ::HGH MX leu2::Gal1p-Sic1(9 m)/ His3p-Gal1/His3p-Gal2/ Gal1p-Gal4 (single copy)* | This study | K20891 | Materials and methods subsection experimental models |
| Strain, strain background (*S. cerevisiae*) | *S. cerevisiae Mat a MATa SCC1-PK9::KanMX rad61Δ::HGH MX smc3(R1008I)::HIS3 scc2-45::NAT MX (L545P D575G) leu2::Gal1p-Sic1(9 m)/ His3p-Gal1/His3p-Gal2/ Gal1p-Gal4(single copy)* | This study | K22294 | Materials and methods subsection experimental models |
| Strain, strain background (*S. cerevisiae*) | *S. cerevisiae Mat a MATa SCC1-PK9::KanMX rad61Δ::HGH MX smc3(R1008I)::HIS3 leu2::Gal1p-Sic1(9 m)/ His3p-Gal1/His3p-Gal2/ Gal1p-Gal4(single copy)* | This study | K22296 | Materials and methods subsection experimental models |
| Strain, strain background (*S. cerevisiae*) | *S. cerevisiae MATa eco1Δ::NatMX4 rad61Δ::HGH MX scc2-45::NAT MX (L545P D575G) Smc3(E199A/R1008I)::HIS3 7.5 kb minichromosome (TRP1 ARS1 CEN1-6KB)* | This study | K22297 | Materials and methods subsection experimental models |
| Strain, strain background (*S. cerevisiae*) | *S. cerevisiae MATa eco1Δ::NatMX4 rad61Δ::HGH MX Smc3(E199A/R1008I):: HIS3 7.5 kb minichromosome (TRP1 ARS1 CEN1-6KB)* | This study | K22298 | Materials and methods subsection experimental models |
| Strain, strain background (*S. cerevisiae*) | *S. cerevisiae MATa scc1(A547C)-PK6::KanMX smc3(E570C,S1043C)::ADE2 smc1(G22C,K639C)::NatMX 2.3 kb Trp1-ARS1-Cen4 plasmid* | *Srinivasan et al. (2018)* | K23889 | Materials and methods subsection experimental models |
| Strain, strain background (*S. cerevisiae*) | *S. cerevisiae MATa scc1(A547C)-PK6::KanMX smc3(E570C,S1043C)::ADE2 smc1(G22C,K639C)::NatMX leu2::Gal1p-Sic1(9 m)/ His3p-Gal1/His3p-Gal2/Gal1p-Gal4 (single copy) 2.3 kb Trp1-ARS1-Cen4 plasmid* | *Srinivasan et al. (2018)* | K23972 | Materials and methods subsection experimental models |
| Strain, strain background (*S. cerevisiae*) | *S. cerevisiae MATa Scc1(A547C)-pk6::KanMX Smc3(E570C,S1043C):: ADE2 Smc1(G22c,K639C):: NatMXscc2-45::natMX (L545P D575G) 2.3 kb Trp1-ARS1-Cen4 plasmid* | *Srinivasan et al. (2018)* | K24267 | Materials and methods subsection experimental models |

*Continued on next page*

*Continued*

| Reagent type (Species) or resource | Designation | Source Or reference | Identifiers | Additional information |
|---|---|---|---|---|
| Strain, strain background (S. cerevisiae) | S. cerevisiae MATa Smc1(G22C,K639C)::NatMX4 Smc3(E570C,S1043C)::ADE2 leu2::Gal-Scc1 (R180E,R268D, A547C)-PK6::LEU2 SCC3 (K404E)-HA3::HIS 2.3 kb Trp1-ARS1-Cen4 plasmid | This study | K24697 | Materials and methods subsection experimental models |
| Strain, strain background (S. cerevisiae) | S. cerevisiae MATa Scc1(A547C)-pk6::KanMX Smc3(E570C,S1043C)::ADE2 Smc1(G22c,K639C)::NatMX scc2-45::natMX (L545P D575G) scc3 (K404E)-HA3::HIS 2.3 kb Trp1-ARS1-Cen4 plasmid | This study | K24709 | Materials and methods subsection experimental models |
| Strain, strain background (S. cerevisiae) | S. cerevisiae MATa Smc1(G22C,K639C)::NatMX4 Smc3(E570C,S1043C)::ADE2 leu2::Gal-Sc1(R180E,R268D, A547C)-PK6::LEU2 scc3 (K404E)-HA3::HIS scc2-45::natMX (L545P D575G) TRP1-ARS1-CEN4 2.3 KB plasmid | This study | K24738 | Materials and methods subsection experimental models |
| Strain, strain background (S. cerevisiae) | S. cerevisiae MATa trp1::smc3(E1155Q)-PK6::TRP1 smc1(D1164E) | This study | K25039 | Materials and methods subsection experimental models |
| Strain, strain background (S. cerevisiae) | S. cerevisiae MATa Scc1(A547C)-pk6::KanMX Smc3(E570C,S1043C)::ADE2 Smc1(G22c,K639C)::NatMX scc2-45::natMX (L545P D575G) 2.3 kb Trp1-ARS1-Cen4 plasmid | This study | K25238 | Materials and methods subsection experimental models |
| Strain, strain background (S. cerevisiae) | S. cerevisiae MATa scc1(A547C)-PK6::KanMX pds5 (s81R)::HIS3 smc3(E570C,S1043C)::ADE2 smc1(G22C,K639C)::NatMX leu2::Gal1p-Sic1(9 m)/ His3p-Gal1/His3p-Gal2/Gal1p-Gal4 (single copy) 2.3 kb Trp1-ARS1-Cen4 plasmid | This study | K25311 | Materials and methods subsection experimental models |
| Strain, strain background (S. cerevisiae) | S. cerevisiae MATa scc1(A547C)-PK6::KanMX pds5 (s81R)::HIS3 smc3(E570C,S1043C)::ADE2 smc1(G22C,K639C)::NatMX scc2-45::natMX (L545P D575G) leu2::Gal1p-Sic1(9 m)/ His3p-Gal1/His3p-Gal2/ Gal1p-Gal4 (single copy) 2.3 kb Trp1-ARS1-Cen4 plasmid | This study | K25312 | Materials and methods subsection experimental models |
| Strain, strain background (S. cerevisiae) | S. cerevisiae MATa scc1(A547C)-PK6::KanMX scc3 (K404e)-HA3::HIS3 smc3(E570C,S1043C)::ADE2 smc1(G22C,K639C)::NatMX leu2::Gal1p-Sic1(9 m)/ His3p-Gal1/His3p-Gal2/ Gal1p-Gal4 (single copy) 2.3 kb Trp1-ARS1-Cen4 plasmid | This study | K25313 | Materials and methods subsection experimental models |

*Continued on next page*

*Continued*

| Reagent type (Species) or resource | Designation | Source Or reference | Identifiers | Additional information |
|---|---|---|---|---|
| Strain, strain background (*S. cerevisiae*) | *S. cerevisiae MATa scc1(A547C)-PK6::KanMX scc3 (K404e)-HA3::HIS3 smc3(E570C,S1043C)::ADE2 smc1(G22C,K639C)::NatMX scc2-45::natMX (L545P D575G) leu2::Gal1p-Sic1(9 m)/ His3p-Gal1/His3p-Gal2/Gal1p-Gal4 (single copy) 2.3 kb Trp1-ARS1-Cen4 plasmid* | This study | K25316 | Materials and methods subsection experimental models |
| Strain, strain background (*S. cerevisiae*) | *S. cerevisiae MATa scc1(A547C)-PK6::KanMX smc3(E570C,S1043C)::ADE2 smc1(G22C,K639C)::NatMX scc2-45::natMX (L545P D575G) cdc28-as1* | This study | K25423 | Materials and methods subsection experimental models |
| Strain, strain background (*S. cerevisiae*) | *S. cerevisiae MATa smc3(E570C,S1043C)::ADE2 smc1(G22C,K639C)::NatMX leu2::Gal-Scc1(R180E,R268D, A547C)-PK6::LEU2 scc3 (K404e)-HA3::HIS3 cdc28-as1 2.3 kb Trp1-ARS1-Cen4 plasmid* | This study | K25437 | Materials and methods subsection experimental models |
| Strain, strain background (*S. cerevisiae*) | *S. cerevisiae MATa smc3(E570C,S1043C)::ADE2 smc1(G22C,K639C)::NatMX leu2::Gal-Scc1(R180E,R268D, A547C)-PK6::LEU2 scc2-45 ::natMX (L545P D575G) scc3 (K404e)-HA3::HIS3 cdc28-as1 2.3 kb Trp1-ARS1-Cen4 plasmid* | This study | K25440 | Materials and methods subsection experimental models |
| Strain, strain background (*S. cerevisiae*) | *S. cerevisiae MATa smc3(E570C,S1043C)::ADE2 smc1(G22C,K639C)::NatMX scc1(A547C)-PK6::KanMX scc2-45::natMX (L545P D575G) eco1Δ::HGH MX scc3 (K404E)-HA3::HIS TRP1-ARS1-CEN4 2.3 KB plasmid* | This study | K25947 | Materials and methods subsection experimental models |
| Strain, strain background (*S. cerevisiae*) | *S. cerevisiae MATa ura::ADH1 promoter-OsTIR1-9myc::URA3 PDS5-Pk3-aid::KanMX4scc2-45 ::NatMX scc3 (K404E)::HA3:: HIS leu2::Gal1p-Sic1(9 m)/ His3p-Gal1/His3p-Gal2/ Gal1p-Gal4::Leu2 (single copy)* | This study | K26414 | Materials and methods subsection experimental models |
| Strain, strain background (*S. cerevisiae*) | *S. cerevisiae MATa ura::ADH1 promoter-OsTIR1-9myc::URA3 PDS5-Pk3-aid::KanMX4 scc3 (K404E)::HA3::HIS leu2::Gal1p-Sic1(9 m)/ His3p-Gal1/His3p-Gal2/ Gal1p-Gal4::Leu2 (single copy)* | This study | K26415 | Materials and methods subsection experimental models |
| Strain, strain background (*S. cerevisiae*) | *S. cerevisiae MATa smc1(D1164E)::HIS3MX SCC1-PK9::KanMX leu2::Gal1p-Sic1(9 m)/ His3p-Gal1/His3p-Gal2/ Gal1p-Gal4::Leu2 (single copy)* | This study | K26765 | Materials and methods subsection experimental models |

*Continued on next page*

Continued

| Reagent type (Species) or resource | Designation | Source Or reference | Identifiers | Additional information |
|---|---|---|---|---|
| Strain, strain background (S. cerevisiae) | S. cerevisiae MATa smc1 (D1164E)::HIS3MX SCC1-PK9::KanMX scc2 -45::natMX (L545P D575G) leu2::Gal1p-Sic1(9 m) /His3p-Gal1/His3p-Gal2/ Gal1p-Gal4::Leu2 (single copy) | This study | K26766 | Materials and methods subsection experimental models |
| Strain, strain background (S. cerevisiae) | S. cerevisiae MATa ura::ADH1 promoter-OsTIR1- 9myc::URA3 scc2-3XmAID::KANMX6 Scc1-PK9::KanMX rad61Δ::HGH MX leu2::Gal1p-Sic1(9 m)/ His3p-Gal1/His3p-Gal2 /Gal1p-Gal4::Leu2 (single copy) | This study | K26831 | Materials and methods subsection experimental models |
| Strain, strain background (S. cerevisiae) | S. cerevisiae MATa scc2-45::natMX (L545P D575G) leu2::Gal1p-Sic1(9 m)/ His3p-Gal1/His3p-Gal2/ Gal1p-Gal4::Leu2 (single copy) ura3::Scc1P-Smc3(E570C) -TEV3-Scc1(A547C)-PK9:: KanMX::URA3 (single integrant,fusion linker: (GGGGS)x8 + TEV3) | This study | K26993 | Materials and methods subsection experimental models |
| Strain, strain background (S. cerevisiae) | S. cerevisiae MATa scc2-45::natMX (L545P D575G) leu2::Gal1p-Sic1(9 m)/ His3p-Gal1/His3p-Gal2/ Gal1p-Gal4::Leu2 (single copy) Smc1(G22C, K639C)::NAT MX ura3::Scc1P-Smc3(E570C)- TEV3-Scc1(A547C)- PK9::KanMX::URA3 (single integrant, fusion linker: (GGGGS)x8 + TEV3) | This study | K26994 | Materials and methods subsection experimental models |
| Strain, strain background (S. cerevisiae) | S. cerevisiae MATa bar1::hisG ura3-1::GAL- OsTIR1-9Myc::URA3 cdc45-AID (CDC45-5xGA-IAA17)::kanMX rad61-3slAA::natNT2 SCC1-PK6::TRP1 scc2-45::natMX (L545P D575G) | This study | K27168 | Materials and methods subsection experimental models |
| Strain, strain background (S. cerevisiae) | S. cerevisiae MATa bar1::hisG ura3-1::GAL-OsTIR1 -9Myc::URA3 cdc45-AID (CDC45-5xGA-IAA17)::kanMX rad61-3slAA::natNT2 SCC1-PK6::TRP1 | This study | K27169 | Materials and methods subsection experimental models |
| Strain, strain background (S. cerevisiae) | S. cerevisiae MATa TET R-GFP::LEU2 TETOs::URA3 pds5 S81R-HA6::HIS (NO HA tag) trp1:MET3p-CDC20::TRP1 | This study | K27443 | Materials and methods subsection experimental models |

*Continued*

| Reagent type (Species) or resource | Designation | Source Or reference | Identifiers | Additional information |
|---|---|---|---|---|
| Strain, strain background (*S. cerevisiae*) | *S. cerevisiae MATa SCC1-PK9::KanMX smc3(T1185M)-HIS3M × 6 scc2-45::natMX (L545P D575G) leu2::Gal1p-Sic1(9 m)/ His3p-Gal1/His3p-Gal2/ Gal1p-Gal4::Leu2 (single copy)* | This study | K27536 | Materials and methods subsection experimental models |
| Strain, strain background (*S. cerevisiae*) | *S. cerevisiae MATa SCC1-PK9::KanMX smc3(T1185M)-HIS3M × 6 leu2::Gal1p-Sic1(9 m)/ His3p-Gal1/His3p-Gal2/ Gal1p-Gal4::Leu2 (single copy)* | This study | K27537 | Materials and methods subsection experimental models |
| Strain, strain background (*S. cerevisiae*) | *S. cerevisiae MATa SCC1-PK9::KanMX rad61Δ::hghMX leu2::Gal1p-Sic1 (9 m)/His3p-Gal1/His3p- Gal2/Gal1p-Gal4::Leu2 (single copy)* | This study | K27569 | Materials and methods subsection experimental models |
| Strain, strain background (*S. cerevisiae*) | *S. cerevisiae MATa SCC1-PK9::KanMX rad61Δ::hghMX scc4-2:NAT leu2::Gal1p-Sic1 (9 m)/His3p-Gal1/His3p -Gal2/Gal1p-Gal4 ::Leu2 (single copy)* | This study | K27570 | Materials and methods subsection experimental models |
| Strain, strain background (*S. cerevisiae*) | *S. cerevisiae MATa scc2-4 TET R-GFP::LEU2 TETOs::URA3 pds5 S81R-HA6::HIS (NO HA tag) trp1:MET3p-CDC20::TRP1* | This study | K27575 | Materials and methods subsection experimental models |
| Strain, strain background (*Spodoptera frugiperda*) | Sf9 insect cells | ThermoFisher | Cat# 11496015 | |
| Antibody | Mouse monoclonal Anti-V5 | BioRad | Cat# MCA1360 | (1:1000) |
| Antibody | Mouse monoclonal Anti Smc3 Acetyl | *Beckouët et al. (2010)* | N/A | (1:1000) |
| Antibody | Rabbit polyclonal Anti Clb2 y-180 | SantaCruz | Sc-9071 | (1:1000) |
| Antibody | Rabbit polyclonal Anti Sic1 FL-284 | SantaCruz | Sc-50441 | (1:1000) |
| Antibody | Mouse monoclonal Anti PGK1 | ThermoFisher Scientific | Cat#459250 | (1:5000) |
| Antibody | Rabbit polyclonal Anti-mini-AID-tag mAb | Kanemaki MT/MBL International | M214-3 | (1:1000) |
| Chemical compound, drug | Acid-washed glass beads | Sigma | Cat# G8722 | |
| Chemical compound, drug | ATP α-$^{32}$P | Hartmann Analytic | Cat# SRP-203 | |
| Chemical compound, drug | Bismaleimidoethane (BMOE) | ThermoFisher | Cat# 22323 | (5 mM) |
| Chemical compound, drug | Complete EDTA free protease inhibitor cocktail | Roche | Cat# 4693132001 | (1:50 ml) |

*Continued on next page*

*Continued*

| Reagent type (Species) or resource | Designation | Source Or reference | Identifiers | Additional information |
|---|---|---|---|---|
| Chemical compound, drug | Dithiothreitol | Fluka | Cat# BP172 | (5 mM) |
| Chemical compound, drug | DMSO | Sigma | Cat# D8418 | |
| Chemical compound, drug | Immobilon Western ECL | Millipore | Cat# WBLKS0500 | |
| Chemical compound, drug | Trisodium citrate | Sigma | Cat# W302600 | |
| Chemical compound, drug | RNase A | Roche | Cat# 10109169001 | |
| Chemical compound, drug | Nocodazole | Sigma | Cat# M1404 | |
| Chemical compound, drug | 1NMPP1 | MerkMillipore | 529581 | |
| Chemical compound, drug | Indole-3-acetic acid (auxin) | Sigma | Cat# I3750-5G-A | |
| Chemical compound, drug | PMSF | Sigma | Cat# 329-98-6 | |
| Chemical compound, drug | Potassium chloride | Sigma | Cat# P5405 | |
| Chemical compound, drug | Proteinase K | Roche | Cat# 03115836001 | |
| Chemical compound, drug | Sodium sulfite | Sigma | Cat# 71988 | |
| Peptide, recombinant protein | α-factor peptide | CRUK Peptide Synthesis Service | | |
| Biological sample (*Saccharomyces cerevisiae*) | *Saccharomyces cerevisiae* cohesin | This study | | Materials and methods subsection Tetramer and Scc2 purification |
| Biological sample (*Saccharomyces cerevisiae*) | *Saccharomyces cerevisiae* Scc2 | This study | | Materials and methods subsection Tetramer and Scc2 purification |
| Commercial assay or kit | AxyPrep Mag PCR Clean up Kit | Appleton Woods Ltd. | Cat# AX402 | |
| Commercial assay or kit | ChIP Clean and Concentrator Kit | Zymo Research | Cat# D5205 | |
| Commercial assay or kit | E-Gel SizeSelect II Agarose Gels, 2% | ThermoFisher | Cat# G661012 | |
| Commercial assay or kit | EnzChek phosphate assay kit | Invitrogen | Cat# E6646 | |
| Commercial assay or kit | HiTrap TALON column | GE Healthcare | Cat# 28-9537-67 | |
| Commercial assay or kit | Library Quantification Kit Ion Torrent Platforms | KAPA Biosystems | Cat# KR0407 | |
| Commercial assay or kit | Microcon YM-100 columns | Sigma | Cat# Z648094 | |

*Continued on next page*

*Continued*

| Reagent type (Species) or resource | Designation | Source Or reference | Identifiers | Additional information |
|---|---|---|---|---|
| Commercial assay or kit | NEBNext Fast DNA library prep set for Ion Torrent | NEB | Cat# E6270L | |
| Commercial assay or kit | NuPAGE 3–8% Tris-Acetate Protein Gels, 1.5 mm, 10-well | ThermoFisher | Cat# EA0378BOX | |
| Commercial assay or kit | NuPAGE 4–12% Bis-Tris Protein Gels, 1.0 mm, 10-well | ThermoFisher | Cat# NP0321BOX | |
| Commercial assay or kit | Prime-it II Random Primer Labelling Kit | Agilent | Cat# 300385 | |
| Commercial assay or kit | Protein G dynabeads | ThermoFisher | Cat# 10003D | |
| Commercial assay or kit | Slide-a-lyzer dialysis units (3.5 kDa) | ThermoFisher | Cat# 66330 | |
| Commercial assay or kit | StrepTrap HP column | GE Healthcare | Cat# 28-9075-48 | |
| Commercial assay or kit | Superdex 200 16/60 GL | GE Healthcare | Cat# 17-1069-01 | |
| Commercial assay or kit | Superose 6 10/300 GL | GE Healthcare | Cat# 17517201 | |
| Commercial assay or kit | TALON Superflow metal affinity resin | Clontech | Cat# 635670 | |
| Software, algorithm | Galaxy platform | *Giardine et al. (2005)* | https://usegalaxy.org | |
| Software, algorithm | FastQC | Galaxy tool version 1.0.0 | https://usegalaxy.org | |
| Software, algorithm | Trim sequences | Galaxy tool version 1.0.0 | https://usegalaxy.org | |
| Software, algorithm | Filter FASTQ | Galaxy tool version 1.0.0 | https://usegalaxy.org | |
| Software, algorithm | Bowtie2 | *Langmead and Salzberg (2012)* Galaxy tool version 0.2 | https://usegalaxy.org | |
| Software, algorithm | Bam to BigWig | Galaxy tool version 0.1.0 | https://usegalaxy.org | |
| Software, algorithm | Samtools | *Li et al. (2009)* | http://samtools. sourceforge.net/ | |
| Software, algorithm | IGB browser | *Nicol et al. (2009)* | http://bioviz.org/igb/ | |
| Software, algorithm | Filter SAM or BAM | *Li et al. (2009)* Galaxy tool version 1.1.0 | https://usegalaxy.org | |
| Software, algorithm | chr_position.py | This study | https://github.com /naomipetela/ nasmythlab-ngs | |
| Software, algorithm | filter.py | This study | https://github.com/ naomipetela/ nasmythlab-ngs | |
| Software, algorithm | bcftools call | *Li et al. (2009)* | | |

## Contact for reagent and resource sharing

Further information and requests for resources and reagents should be directed to and will be fulfilled by the lead contact Kim Nasmyth (ashley.nasmyth@bioch.ox.ac.uk).

## Yeast cell culture

All strains are derivatives of W303 (K699). Strain numbers and relevant genotypes of the strains used are listed in the Key Resource Table. Cells were cultured at 25°C in YEP medium with 2% glucose unless stated otherwise. To arrest the cells in G1, α-factor was added to a final concentration of 2 mg/L, every 30 min for 2.5 hr. Cells were released from G1 arrest by filtration wherein cells were captured on 1.2 μm filtration paper (Whatman GE Healthcare), washed with 1 L YEPD and resuspended in the appropriate fresh media. To inactivate Scc2 (scc2-45: temperature sensitive allele), fresh media was pre-warmed prior to filtration.

To arrest cells in G2, nocodazole (Sigma) was added to the fresh media to a final concentration of 10 μg/mL and cells were incubated until the synchronization was achieved (>95% large-budded cells).

Cells were arrested in late G1 by galactose-induced overexpression of a non-degradable mutant of the Sic1 protein (mutation of 9 residues phosphorylated by Cdk1). To achieve this, cells were grown in YEP supplemented with 2% raffinose and arrested in G1 as described above. 1 hr before release from G1 arrest, galactose was added to 2% of the final concentration. Cells were released into YEPD as described above, and incubated for 60 min at 25°C.

For auxin-induced degradation of scc2-3XmAID and Pds5-AID in late G1, cells were arrested in late G1 as described above. 1 hr prior to release from alpha-factor arrest, auxin (indole-3-acetic acid sodium salt; Sigma) was added to a final concentration of 3 mM. Cells were released from G1 arrest into YEPD medium containing 3 mM auxin.

For auxin-induced degradation of scc2-3XmAID and Pds5-AID in G2, cells were arrested in G2 as described above. Once >95% cells were arrested in G2, auxin (indole-3-acetic acid sodium salt; Sigma) was added to a final concentration of 5 mM and incubated for 60 min.

For depletion of Cdc45-AID, cells carrying cdc45-AID and Gal1p-OsTIR1 were arrested in G1 with alpha factor in YEP Raffinose medium. 1 hr prior to release from the G1 arrest, (2% final) galactose and auxin 3 mM were added. Cells were released from the G1 arrest into YEPD medium containing 3 mM auxin.

To inhibit CDK1 cdc28-as1 cells were arrested in G2 with nocodazole (Sigma) until synchronization was achieved (>95% large-budded cells) at 25°C. Subsequently, 1NMPP1 (5μM final) was added and the cultures incubated for 60 min at 25°C.

To induce re-replication, Cultures where CDK1 was inhibited (as described above) were filtered and washed with 1 l fresh YEPD medium, the cells were resuspended in fresh YEPD medium containing nocodazole and wither 1NMPP1 or DMSO and incubated for further 90–120 min.

## In vivo chemical crosslinking (For western blotting and minichromosome IP)

Strains were grown in YEPD at 25°C to $OD_{600nm}$ = 0.5–0.6. 12 OD units were washed in ice-cold PBS and re-suspended in 1 mL ice-cold PBS. The suspensions were then split into 2 × 500 μL and 20.8 μL BMOE (stock: 125 mM in DMSO, 5 mM final) or DMSO was added for 6 min on ice. Cells were washed with 2 × 2 mL ice-cold PBS containing 5 mM DTT, resuspended in 500 μL lysis buffer (25 mM Hepes pH 8.0, 50 mM KCl, 50 mM $MgSO_4$, 10 mM trisodium citrate, 25 mM sodium sulfite, 0.25% triton-X, freshly supplemented with Roche Complete Protease Inhibitors (2X) and PMSF (1 mM), lysed in a FastPrep-24 (MP Biomedicals) for 3 × 1 min at 6.5 m/s with 500 μl of acid-washed glass beads (425–600 μm, Sigma) and lysates cleared (5 min, 12 kg). Protein concentrations were adjusted after Bradford assay and cohesin immuno-precipitated:

for western blotting: using Anti-HA high affinity matrix (Roche).

for minichromosome IP: using anti-PK antibody (AbD Serotec, 1 hr, 4°C) and protein G dynabeads (1 hr, 4°C, with rotation).

Beads were washed with 3 × 1 mL lysis buffer, resuspended in 50 μl 2x sample buffer, incubated at 95°C for 5 min and the supernatant loaded onto either 3–8% Tris-acetate or 4–12% Bis-Tris gradient gels (Life Technologies).

## Minichromosome IP

Strains containing a 2.3 kb circular minichromosome harbouring the *TRP1* gene were grown over-night in –TRP medium at 25°C and sub-cultured in YEPD medium for exponential growth ($OD_{600nm}$ = 0.6). 30 OD units were washed in ice-cold PBS and processed for in vivo crosslinking as described above with the following modification: after cohesin immuno-precipitation protein G dynabeads were washed with 2 × 1 ml lysis buffer, resuspended in 30 μl 1% SDS with DNA loading dye, incubated at 65°C for 4 min and the supernatant run on a 0.8% agarose gel containing ethidium bromide (1.4 V/cm, 22 hr, 4°C). After Southern blotting using alkaline transfer, bands were detected using a 32 P labeled TRP1 probe.

## SDS gel electrophoresis and western blotting

Whole cell lysates were resolved in NuPAGE 3–8% or 4–12% gradient gels (ThermoFisher Scientific) and transferred onto PVDF membranes using the Trans-blot Turbo transfer system (BioRad). For visualization, the membrane was incubated with Immobilon Western Chemiluminescent HRP substrate (Millipore) before detection using an ODYSSEY Fc Imaging System (LI-COR).

## Differential sedimentation of cleared cell lysates

Cells were harvested at 3500 rpm in Heraeus Multifuge, and pellets were washed twice with cold H2O, resuspended in 100 mM Tris HCl (pH 9.4), 10 mM DTT, and 10 μg/ml nocodazole, and incubated for 20 min on ice. Cells were washed with ice-cold H2O, resuspended in spheroplasting buffer (1 M sorbitol, 50 mM Tris HCl [pH 7.5], 1 mM CaCl2, 1 mM MgCl2, 10 μg/ml nocodazole, 350 U lyticase L4025-Sigma) and incubated 30 min on an orbital platform at 4°C. Spheroplasts were sedimented in a Beckman Coulter JA25.50 at 6000 rpm for 6 min, gently washed with 1 M sorbitol, transferred to 1.5 ml tubes, and sedimented for 1 min at 1500 rcf and 4°C. Pellets were resuspended in 200 μl cold 0.4 M sorbitol and lysed on ice for 30 min by the addition of 700 μl lysis buffer (25 mM HEPES/KOH [pH 8], 50 mM KCl, 10 mM MgSO4, 0.25% Triton X-100, 1 mM PMSF, 3 mM DTT, 1 × complete EDTA-free protease inhibitors), supplemented with 100 μg/ml RNase A and 300 mM NaCl. Cell extracts were obtained by spinning the lysed spheroplasts at 12,000 rcf and 4°C for 5 min.

Cleared lysates (450 μl) were loaded on sucrose gradients prepared in Biocomp gradient station and sedimented in SW41 rotor (Beckman Optima L-100 XP Preparative Ultracentrifuge) at 18,000 rpm for 4 hr. Gradients were fractionated using Gilson FC203B fractionator, collecting 15 drops/fraction.

## Sister chromatid cohesion assay

Cells were grown in −Met medium at 25°C were diluted to OD600 = 0.075 grown to OD600 = 0.15. Cultures were arrested in G1 with alpha-factor and released from G1 arrest into YEPD +2 mM methionine medium at either 25°C or 35.5°C. Samples were drawn every 30 min (up to 150 mins) and fixed with cold 50% ethanol and stored at 4°C. The fixed cells were sonicated for 10 s at 40% power and embedded into a 2% agarose patch on coverslips. GFP fluorescence was observed with a Zeiss Axio Imager.Z1 microscope (63 × objective, NA = 1.40) equipped with a coolSNAP HQ camera. For each experimental condition, at least 100 cells were scored for GFP dots. And each experiment repeated three times.

## FACS analysis

Approximately, $0.5 \times 10^7$ cells were sedimented at 13 k rcf for 30 s, and pellets were fixed with 1 ml 50% ethanol and stored at 5°C. The fixed cells were spun at 6 k rcf and the pellets resuspended in 1 ml 50 mM Tris-HCl (pH 7.5)+20 μl of 10 mg/ml RNaseA and incubated with shaking at 37°C overnight. Cells were pelleted and resuspended in 500 μl PI buffer (200 mM Tris-HCl [pH 7.5], 211 mM NaCl, 78 mM MgCl2) and propidium iodide was added at 50 μg/ml final concentration. Samples were sonicated for 5 s at 40% power and 50–100 μl was diluted into 1 ml 50 mM Tris-HCl (pH 7.5) and read with a Becton Dickinson FACSCalibur, ensuring 30,000 events per sample.

## Calibrated ChIP-sequencing

Cells were grown exponentially to $OD_{600}$ = 0.5 and the required cell cycle stage where necessary. 15 $OD_{600nm}$ units of *S. cerevisiae* cells were then mixed with 5 $OD_{600nm}$ units of *C. glabrata* to a total

volume of 45 mL and fixed with 4 mL of fixative (50 mM Tris-HCl, pH 8.0; 100 mM NaCl; 0.5 mM EGTA; 1 mM EDTA; 30% (v/v) formaldehyde) for 30 min at room temperature (RT) with rotation.

The fixative was quenched with 2 mL of 2.5 M glycine (RT, 5 min with rotation). The cells were then harvested by centrifugation at 3,500 rpm for 3 min and washed with ice-cold PBS. The cells were then resuspended in 300 µL of ChIP lysis buffer (50 mM Hepes-KOH, pH 8.0; 140 mM NaCl; 1 mM EDTA; 1% (v/v) Triton X-100; 0.1% (w/v) sodium deoxycholate; 1 mM PMSF; 2X Complete protease inhibitor cocktail (Roche)) and an equal amount of acid-washed glass beads (425–600 µm, Sigma) added before cells were lysed using a FastPrep−24 benchtop homogeniser (M.P. Biomedicals) at 4˚ C (3 × 60 s at 6.5 m/s or until >90% of the cells were lysed as confirmed by microscopy).

The soluble fraction was isolated by centrifugation at 2,000 rpm for 3 min then sonicated using a bioruptor (Diagenode) for 30 min in bursts of 30 s on/30 s off at high level in a 4˚C water bath to produce sheared chromatin with a size range of 200–1,000 bp. After sonication the samples were centrifuged at 13,200 rpm at 4˚C for 20 min and the supernatant was transferred into 700 µL of ChIP lysis buffer. 30 µL of protein G Dynabeads (Invitrogen) were added, and the samples were pre-cleared for 1 hr at 4˚C. 80 µL of the supernatant was removed (termed 'whole cell extract [WCE] sample') and 5 µg of antibody (anti-PK (Bio-Rad) or anti-HA (Roche)) was added to the remaining supernatant which was then incubated overnight at 4˚C. 50 µL of protein G Dynabeads were then added and incubated at 4˚C for 2 hr before washing 2x with ChIP lysis buffer, 3x with high salt ChIP lysis buffer (50 mM Hepes-KOH, pH 8.0; 500 mM NaCl; 1 mM EDTA; 1% (v/v) Triton X-100; 0.1% (w/v) sodium deoxycholate; 1 mM PMSF), 2x with ChIP wash buffer (10 mM Tris-HCl, pH 8.0; 0.25 M LiCl; 0.5 % NP-40; 0.5% sodium deoxycholate; 1 mM EDTA; 1 mM PMSF) and 1x with TE pH7.5. The immunoprecipitated chromatin was then eluted by incubation in 120 µL TES buffer (50 mM Tris-HCl, pH 8.0; 10 mM EDTA; 1% SDS) for 15 min at 65˚C and the collected supernatant termed 'IP sample'. The WCE samples were mixed with 40 µL of TES3 buffer (50 mM Tris-HCl, pH 8.0; 10 mM EDTA; 3% SDS), and all samples were de-crosslinked by incubation at 65˚C overnight. RNA was degraded by incubation with 2 µL RNase A (10 mg/mL; Roche) for 1 hr at 37˚C and protein was removed by incubation with 10 µL of proteinase K (18 mg/mL; Roche) for 2 hr at 65˚C. DNA was purified using ChIP DNA Clean and Concentrator kit (Zymo Research).

## Extraction of yeast DNA for deep sequencing

Cultures were grown to exponential phase (OD600 = 0.5). 12.5 OD600 units were then collected and diluted to a final volume of 45 mL before fixation as described in the protocol for ChIP-seq. The samples were treated as specified in the ChIP-seq protocol up to the completion of the sonication step whereby 80 µL of the samples were carried forward and treated as WCE samples.

## Preparation of sequencing libraries

Sequencing libraries were prepared using NEBNext Fast DNA Library Prep Set for Ion Torrent Kit (New England Biolabs) according to the manufacturer's instructions. Briefly, 10–100 ng of fragmented DNA was converted to blunt ends by end repair before ligation of the Ion Xpress Barcode Adaptors. Fragments of 300 bp were then selected using E-Gel SizeSelect2% agarose gels (Life Technologies) and amplified with 6–8 PCR cycles. The DNA concentration was then determined by qPCR using Ion Torrent DNA standards (Kapa Biosystems) as a reference. 12–16 libraries with different barcodes could then be pooled together to a final concentration of 350 pM and loaded onto the Ion PI V3 Chip (Life Technologies) using the Ion Chef (Life Technologies). Sequencing was then completed on the Ion Torrent Proton (Life Technologies), typically producing 6–10 million reads per library with an average read length of 190 bp.

## Data analysis, alignment and production of BigWigs

Unless otherwise specified, data analysis was performed on the Galaxy platform. Quality of reads was assessed using FastQC (Galaxy tool version 1.0.0) and trimmed as required using 'trim sequences' (Galaxy tool version 1.0.0). Generally, this involved removing the first 10 bases and any bases after the 200[th], but trimming more or fewer bases may be required to ensure the removal of kmers and that the per-base sequence content is equal across the reads. Reads shorter than 50 bp were removed using Filter FASTQ (Galaxy tool version 1.0.0, minimum size: 50, maximum size: 0, minimum quality: 0, maximum quality: 0, maximum number of bases allowed outside of quality range: 0,

paired end data: false) and the remaining reads aligned to the necessary genome(s) using Bowtie2 (Galaxy tool version 0.2) with the default (–sensitive) parameters (mate paired: single-end, write unaligned reads to separate file: true, reference genome: SacCer3 or CanGla, specify read group: false, parameter settings: full parameter list, type of alignment: end to end, preset option: sensitive, disallow gaps within $n$-positions of read: 4, trim $n$-bases from 5' of each read: 0, number of reads to be aligned: 0, strand directions: both, log mapping time: false).

To generate alignments of reads that uniquely align to the *S. cerevisiae* genome, the reads were first aligned to the *C. glabrata* (CBS138, genolevures) genome with the unaligned reads saved as a separate file. These reads that could not be aligned to the *C. glabrata* genome were then aligned to the *S. cerevisiae* (sacCer3, SGD) genome and the resulting BAM file converted to BigWig (Galaxy tool version 0.1.0) for visualisation. Similarly, this process was done with the order of genomes reversed to produce alignments of reads that uniquely align to *C. glabrata*.

## Visualisation of ChIP-seq profiles

The resulting BigWigs were visualised using the IGB browser. To normalise the data to show quantitative ChIP signal the track was multiplied by the samples' occupancy ratio (OR) and normalised to 1 million reads using the graph multiply function. In order to calculate the average occupancy at each base pair up to 60 kb around all 16 centromeres, the BAM file that contains reads uniquely aligning to *S. cerevisiae* was separated into files for each chromosome using 'Filter SAM or BAM' (Galaxy tool version 1.1.0). A pileup of each chromosome was then obtained using samtools mpileup (Galaxy tool version 0.0.1) (source for reference list: locally cached, reference genome: SacCer3, genotype likelihood computation: false, advanced options: basic). These files were then amended using our own script (chr_position.py) to assign all unrepresented genome positions a value of 0. Each pileup was then filtered using another in-house script (filter.py) to obtain the number of reads at each base pair within up to 60 kb intervals either side of the centromeric CDEIII elements of each chromosome. The number of reads covering each site as one successively moves away from these CDEIII elements could then be averaged across all 16 chromosomes and calibrated by multiplying by the samples' OR and normalizing to 1 million reads.

## Cohesin tetramer and Scc2 purification

All versions of the cohesin complexes purified bear a twin StrepII tag on the Scc1 kleisin. This is the same for the Scc2 construct used in this study except the later bears a single Strep-II tag. Typically 500 ml of SF-9 insect cells were grown to ~3 million/ml and infected with the appropriate baculovirus stock in a 1/100 dilution. Infection was monitored daily and cells harvested when lethality (assayed by the trypan blue test) reached no more than 70–80%. Cell pellets were then frozen in liquid nitrogen and stored at 80°C. Upon thawing, the pellets were suspended in a final volume of ~65–70 ml with Buffer A (final concentrations of: 25 mM HEPES pH 8.0, NaCl 150 mM, TCEP-HCl 1 mM and Glycerol 10%) and the suspension was immediately supplemented with two dissolved tablets of Roche Complete Protease (EDTA-free), 75 µg of RNase I and 7 µl of DNaseI (Roche, of 10 U/µl stock). The cells were then sonicated at 80% amplitude for 5 s/burst/35 ml of suspension using a Sonics Vibra-Cell (3 mm microtip). In total, 12 bursts were given for every 35 ml half of the 70 ml suspension (the sonication was always performed in ethanolised ice). A spin at 235,000 x g (45,000 rpm on a Ti45 fixed angle rotor) followed for 45 mins following addition of PMSF to 1 mM final concentration. The isolated cleared extract was supplemented with 2 mM EDTA and was then used to load a 2 × 5 ml StrepTrap HP (Fisher Scientific) column at 1 ml/min in an ÄKTA Purifier 100. Wash with Buffer A at 1 ml/min to the point of ΔAU$_{280nm}$~0 and protein elution ensued using Buffer A + 20 mM desthiobiotin (Fisher Scientific) at 1 ml/min. Peak fractions were analysed using SDS-PAGE and were further purified in a Superose 6 Increase 10/300 (VWR) using Buffer A as running buffer (free of EDTA/PMSF). The resulting peaks were again analysed using SDS-PAGE and the concentration was determined in Nanodrop using A280. Protein was aliquoted and stocked typically in concentrations ranging from 1 to 3 mg/ml.

## ATPase assay

ATPase activity was measured by using the EnzChek phosphate assay kit (Invitrogen) by following the protocol as provided. Cohesin tetramer (Smc1, Smc3, Scc1 and Scc3; final concentration: 50 nM,

final NaCl concentration: 50 mM) was added together with a 40 bp long double stranded DNA (700 nM). The reaction was started with addition of ATP to a final concentration of 1.3 mM (final reaction volume: 150 µl). After completion, a fraction of each reaction was run on SDS-PAGE and the gel stained with coomassie brilliant blue in order to test that the complexes were intact throughout the experiment and that equal amounts were used when testing various mutants and conditions.

## Quantification and statistical analysis

### ATPase assay

ATPase activity was measured by recording absorption at 360 nm every 30 s for 90 min using a PHERAstar FS. $\Delta$AU at 360 nm was translated to $P_i$ release using an equation derived by a standard curve of $KH_2PO_4$ (EnzChek kit). Rates were calculated from the slope of the linear phase (first 10 min). At least two independent biological experiments were performed for each experiment, means and standard deviations are reported for every experiment.

## Data and software availability

### Scripts

All scripts written for this analysis method are available to download from https://github.com/naomi-petela/nasmythlab-ngs (*Petela, 2019*; copy archived at https://github.com/elifesciences-publica-tions/nasmythlab-ngs).

*Chr_position.py* takes mpileups for *S. cerevisiae* chromosomes and fills in gaps, with each position in the chromosome added given a read depth of 0.

*Filter60.py* reads the files produced by Chr_position.py and takes the read depth for all positions 60 kb either side of the CDEIII for all chromosomes, produces an average for each position and multiples it by the OR. The OR should be derived from the reads aligned in the appropriate bam files (*Hu et al., 2015*).

## Calibrated ChIP-seq data

The calibrated ChIP-seq data (raw and analysed data) have been deposited on GEO under accession number GSE132221.

## Acknowledgements

Maria Demidova conducted initial experiments that this study expanded on. We are grateful to Tomo Tanaka and Seiji Tanaka for supplying reagents. We thank all members of the Nasmyth group for valuable discussions, technical assistance and critical reading of the manuscript. This work was funded by the Wellcome Trust Senior Investigator Award, Grant Ref 107935/Z/15/Z and Cancer Research UK Programme Grant, Grant Ref 26747 to KN. BH is funded by (202062/Z/16/Z).

## Additional information

### Funding

| Funder | Grant reference number | Author |
| --- | --- | --- |
| Wellcome | 107935/Z/15/Z | Kim A Nasmyth |
| Cancer Research UK | 26747 | Kim A Nasmyth |
| Wellcome | 202062/Z/16/Z | Bin Hu |

The funders had no role in study design, data collection and interpretation, or the decision to submit the work for publication.

### Author contributions

Madhusudhan Srinivasan, Conceptualization, Data curation, Formal analysis, Supervision, Validation, Investigation, Visualization, Methodology, Writing—original draft, Project administration, Writing—review and editing; Naomi J Petela, Data curation, Formal analysis, Investigation, Methodology; Johanna C Scheinost, Formal analysis, Investigation, Methodology; James Collier, Resources,

Investigation, Methodology; Menelaos Voulgaris, Investigation, Methodology; Maurici B Roig, Resources, Investigation; Frederic Beckouët, Formal analysis, Validation, Investigation; Bin Hu, Conceptualization, Formal analysis, Funding acquisition, Validation, Investigation; Kim A Nasmyth, Conceptualization, Supervision, Funding acquisition, Writing—original draft, Writing—review and editing

## Author ORCIDs

Madhusudhan Srinivasan (iD) https://orcid.org/0000-0001-5676-4219

Kim A Nasmyth (iD) https://orcid.org/0000-0001-7030-4403

## Decision letter and Author response

Decision letter https://doi.org/10.7554/eLife.44736.018
Author response https://doi.org/10.7554/eLife.44736.019

# Additional files

## Data availability

The sequencing data have been deposited in GEO under the accession number GSE132221.

The following dataset was generated:

| Author(s) | Year | Dataset title | Dataset URL | Database and Identifier |
|---|---|---|---|---|
| Naomi J Petela, Kim A Nasmyth | 2019 | Scc2 counteracts a Wapl-independent mechanism that releases cohesin from chromosomes during G1 | https://www.ncbi.nlm.nih.gov/geo/query/acc.cgi?acc=GSE132221 | NCBI Gene Expression Omnibus, GSE132221 |

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
