## [Decision Letter]

Thank you for submitting your article "Scc2 counteracts a Wapl-independent activity that releases cohesin from chromosomes in G1" for consideration by *eLife*. Your article has been reviewed by three peer reviewers, including Bruce Stillman as the Reviewing Editor and Reviewer #1, and the evaluation has been overseen by Kevin Struhl as the Senior Editor. The following individual involved in review of your submission has agreed to reveal their identity: Hongtao Yu (Reviewer #2).

The reviewers have discussed the reviews with one another and the Reviewing Editor has drafted this decision to help you prepare a revised submission.

The paper by Srinivasan et al. addresses the role of Scc2, a component of the cohesion of sister chromatids in cells. Previous studies have demonstrated that Scc2 is required for cohesin loading, and that there is a single pathway involving Wap1 for cohesin unloading. The current paper suggests that Scc2 is also required for maintenance of cohesin binding to chromosomes in G1 phase by preventing release in a Wapl-independent manner. The authors show that this release mechanism does require loop opening of the Smc3/Scc1 interface and the Smc1 ATPase activity. The studies raise key issues about the role of Wapl in release. The Scc2 prevention of the Wapl-independent release is cell cycle regulated and is affected by Cyclin-Cdk1 activity. The authors also suggest that once cohesin is loaded (for example in G2 and inherited into a G1-like state), Scc2 is no longer required for cohesion in a subsequent S phase. This may well be the case, but a key finding needs attention (see major point below).

The paper certainly raises a number of interesting points about the control of both loading and removing cohesin and the roles of key players such as Scc2 and Wapl. However, as it is currently presented, the manuscript is only accessible to a few experts in the field. For the general readership, following even the most basic experiments is difficult to read. A revised paper would be suitable for publication in *eLife*.

Essential revisions:

1) The manuscript tends to disregard some major effects of the mutants used, which the authors deem less relevant. For example, cohesin in the *scc2-45* mutant has already lost the vast majority of its chromosomal association even at permissive temperature (Figure 1B, D and Figure 3—figure supplement 1D). The Smc3-Scc1 fusion by itself renders cells temperature-sensitive. Furthermore, the authors use different mutant alleles for different experiments without apparent reason (e.g. *scc2-45* or *scc2-4, scc3K404E* or *pds5S81R*). We therefore consider that the authors:

a) show absolute ChIP-seq binding profiles for each experiment (as in Figure 1D) instead of relative binding profiles (as, for example, in Figure 1E-H), which will reveal effects of the different mutations on cohesin binding in comparison to 'real' wild-type cells. This would, for example, reveal whether fusion of Scc2 to the AID tag reduces cohesin binding in the absence of auxin in a similar manner as the *scc2-45* mutation does.

b) probe for the steady-state levels of mutant proteins under permissive and ts conditions in each experiment (e.g. Figure 1B-E, 2A: Scc2 and Scc4, Figure 2B: Pds5 and Scc2, Figure 2C: Smc3-Scc1 and Scc2, Figure 6E and G: Scc2). This is important since, for example, Scc4 is a constitutive binding partner of Scc2 and has been reported to be required for the stability the latter (Watrin et al., 2006). One would expect that *scc4-4* inactivation would indirectly reduce Scc2 levels, which, however, doesn't seem to be obvious from ChIP-seq profiles (Figure 2A).

2) At this point, the evidence that Wapl-independent cohesin release also occurs through opening of the Smc3-Scc1 interface is very limited and should be substantiated further with assays that directly probe for Scc1-N release (such as performed previously by the same group; Beckouet et al., 2016). This is particularly important, since half of the Smc3-Scc1 fusion protein is lost from pericentromeric regions after the temperature shift regardless of *scc2-4*/SCC2 status, which raises concerns about the interpretation of the fusion protein experiments.

3) The authors introduce a new mutation in Smc3 (R1008I) that allows the maintenance of *scc2-45* mutants in the absence of Wapl and note that the mechanistic effect of this mutation will be presented elsewhere. Since many conclusions are based on this mutant combination, it will be essential that the authors explain the effect of this mutation. Alternatively, the authors could use the SCC2-3XmAID degron version for most experiments, which doesn't depend on the *smc3R1008I* background and, unlike the *scc2-45* mutant, might not have a cohesin loading defect in the absence of auxin (see comment 1a).

4) The conclusion that Scc2 inactivation has 'only a modest effect on *smc3T1185M*' doesn't seem to match the massive reduction of cohesin at centromeric and pericentric regions (Figure 2E). The same is true for Scc2 inactivation in *scc3K404E* (Figure 4B) or *cdc45* (Figure 4E) cells. The authors seem to largely play down these differences at centromeres and pericentromeres and instead focus their discussion on cohesin at chromosome arm regions (e.g. '…at least along chromosome arms, which represents the majority of chromosomal cohesin'). However, Figure 1D clearly shows that most cohesin is present at centromeric and pericentromeric regions. The differences observed in these regions upon Scc2 inactivation might very well be significant and cannot be explained by a simple release mechanism.

5) Figure 6E and subsection “Scc2 is not required during S phase to establish sister chromatid cohesion”. Is it possible that the CM produced in the previous G2 period are then partially replicated and the CD that appear are a result of concatenation of the DNA rather than cohesion? They would still have cohesin associated with the CM and a second replicated DNA could remain associated by catenation of the DNA. In fact, the authors note the incomplete replication (Figure 6C and Figure 7—figure supplement 1B).

---

## [Author Response]

Essential revisions:1) The manuscript tends to disregard some major effects of the mutants used, which the authors deem less relevant. For example, cohesin in the scc2-45 mutant has already lost the vast majority of its chromosomal association even at permissive temperature (Figure 1B, D and Figure 3—figure supplement 1D). The Smc3-Scc1 fusion by itself renders cells temperature-sensitive. Furthermore, the authors use different mutant alleles for different experiments without apparent reason (e.g. scc2-45 or scc2-4, scc3K404E or pds5S81R). We therefore consider that the authors:a) show absolute ChIP-seq binding profiles for each experiment (as in Figure 1D) instead of relative binding profiles (as, for example, in Figure 1E-H), which will reveal effects of the different mutations on cohesin binding in comparison to 'real' wild-type cells. This would, for example, reveal whether fusion of Scc2 to the AID tag reduces cohesin binding in the absence of auxin in a similar manner as the scc2-45 mutation does.

We have shown the absolute profiles for the experiments described currently in Figure 2. The AID tag affects Scc2’s ability to load even without auxin (Figure 2E and F) as well as altering Wapl-independent release (Figure 4F). *scc2-45* is also partially defective at its permissive temperature. This is not surprising since almost all conditional mutants are defective under so called permissive conditions. We did not point it out in the previous version of the manuscript as we do not believe it compromises their utility. By providing the absolute profiles, the revised manuscript makes this point clearer without readers having to analyse the raw data.

b) probe for the steady-state levels of mutant proteins under permissive and ts conditions in each experiment (e.g. Figure 1B-E, 2A: Scc2 and Scc4, Figure 2B: Pds5 and Scc2, Figure 2C: Smc3-Scc1 and Scc2, Figure 6E and G: Scc2). This is important since, for example, Scc4 is a constitutive binding partner of Scc2 and has been reported to be required for the stability the latter (Watrin et al., 2006). One would expect that scc4-4 inactivation would indirectly reduce Scc2 levels, which, however, doesn't seem to be obvious from ChIP-seq profiles (Figure 2A).

We have probed for steady state levels of Scc1, Smc3-Scc1 fusion protein +/- Scc2 in the relevant experiments. We cannot probe the steady-state levels and the fate of *scc2-45* protein upon temperature shift. It seems irrelevant whether or not Scc2-45 protein disappears or not upon shift to 37°C. The purpose of the shift is to inactivate the protein. Whether this is accompanied by degradation is not really relevant. However, we have good reason to believe that the mutant protein is not destroyed by the temperature shift as unpublished experiments show that re-loading can be rapidly induced by returning cells to the permissive temperature.

Under the conditions we use in late G1 cells, Scc4 inactivation does not affect Scc2 levels. See Figure 3—figure supplement 1C. This is now mentioned and its importance is mentioned in the revised manuscript.

2) At this point, the evidence that Wapl-independent cohesin release also occurs through opening of the Smc3-Scc1 interface is very limited and should be substantiated further with assays that directly probe for Scc1-N release (such as performed previously by the same group; Beckouet et al., 2016). This is particularly important, since half of the Smc3-Scc1 fusion protein is lost from pericentromeric regions after the temperature shift regardless of scc2-4/SCC2 status, which raises concerns about the interpretation of the fusion protein experiments.

Though we do not think that there is any ambiguity as to the interpretation of the *scc2-45* shift experiments with the Smc3-Scc1 fusion, the suggestion by the reviewer to measure the effect of Scc2 depletion on NScc1-Smc3 association is an excellent one, which we had also contemplated. Using in vivo crosslinking, we now show that depletion of Scc2 in a wapl deletion strain releases the NScc1 that remains associated with Smc3. Please see Figure 4D, E and F.

3) The authors introduce a new mutation in Smc3 (R1008I) that allows the maintenance of scc2-45 mutants in the absence of Wapl and note that the mechanistic effect of this mutation will be presented elsewhere. Since many conclusions are based on this mutant combination, it will be essential that the authors explain the effect of this mutation. Alternatively, the authors could use the SCC2-3XmAID degron version for most experiments, which doesn't depend on the smc3R1008I background and, unlike the scc2-45 mutant, might not have a cohesin loading defect in the absence of auxin (see comment 1a).

There has been a slight misunderstanding. We have used the *smc3R1008I* mutant only in one experiment described currently in Figure 2C and D. This was done to show the G1 release in *scc2-45 wpl1* deletion double mutant strains. The mutant has not been used in any other experiment described. We currently do not understand how R1008I suppresses the growth defect caused by *wpl1* deletion in *scc2-45* cells. The key point is that it does not do so by restoring cohesin release that is abolished by *wpl1* deletion. We use it as a tool to address a question. We have modified the statement to explicitly state that we do not understand how R1008I works and that it was used as a tool in just the one experiment. As already pointed out, the Scc2-3XAID is not a particularly useful alternative solution as this allele like *scc2-45* is partially defective in loading even under permissive conditions. The *scc2-45* allele has the huge advantage that its inactivation is clearly vary rapid indeed.

4) The conclusion that Scc2 inactivation has 'only a modest effect on smc3T1185M' doesn't seem to match the massive reduction of cohesin at centromeric and pericentric regions (Figure 2E). The same is true for Scc2 inactivation in scc3K404E (Figure 4B) or cdc45 (Figure 4E) cells. The authors seem to largely play down these differences at centromeres and pericentromeres and instead focus their discussion on cohesin at chromosome arm regions (e.g. '…at least along chromosome arms, which represents the majority of chromosomal cohesin'). However, Figure 1D clearly shows that most cohesin is present at centromeric and pericentromeric regions. The differences observed in these regions upon Scc2 inactivation might very well be significant and cannot be explained by a simple release mechanism.

We play down the effect on peri-centric cohesin for the very good reason that it represents only a minor fraction of the chromosomal cohesin pool. In the case of chromosome IV it represents less than 10%. We currently have no idea why we see differential effects on peri-centric and arm cohesin.

5) Figure 6E and subsection “Scc2 is not required during S phase to establish sister chromatid cohesion”. Is it possible that the CM produced in the previous G2 period are then partially replicated and the CD that appear are a result of concatenation of the DNA rather than cohesion? They would still have cohesin associated with the CM and a second replicated DNA could remain associated by catenation of the DNA. In fact, the authors note the incomplete replication (Figure 6C and Figure 7—figure supplement 1B).

This is an excellent point! We have addressed this and presented the data in Figure 7—figure supplement 1D along with a cartoon. We have excluded the possibility that DNA catenation was holding the re-replicated sister DNAs by showing that the SDS resistant CDs produced by re-replication are dependent on covalent circularization of cohesin with BMOE cross-linking. If under the same conditions cohesin remained un-crosslinked, no SDS resistant species other than the supercoiled min-chromosome monomer was detected. This rules out DNA catenation as a factor.